# Roles for mycobacterial DinB2 in frameshift and substitution mutagenesis

**Pierre Dupuy[1†], Shreya Ghosh[2†], Allison Fay[1], Oyindamola Adefisayo[1,3], Richa Gupta[1], Stewart Shuman[2]\*, Michael S Glickman[1,3]\***

[1]Immunology Program, Sloan Kettering Institute, New York, United States; [2]Molecular Biology Program, Sloan Kettering Institute, New York, United States; [3]Immunology and Microbial Pathogenesis Graduate Program, Weill Cornell Graduate School, New York, United States

**Abstract** Translesion synthesis by translesion polymerases is a conserved mechanism of DNA damage tolerance. In bacteria, DinB enzymes are the widely distributed promutagenic translesion polymerases. The role of DinBs in mycobacterial mutagenesis was unclear until recent studies revealed a role for mycobacterial DinB1 in substitution and frameshift mutagenesis, overlapping with that of translesion polymerase DnaE2. *Mycobacterium smegmatis* encodes two additional DinBs (DinB2 and DinB3) and *Mycobacterium tuberculosis* encodes DinB2, but the roles of these polymerases in mycobacterial damage tolerance and mutagenesis is unknown. The biochemical properties of DinB2, including facile utilization of ribonucleotides and 8-oxo-guanine, suggest that DinB2 could be a promutagenic polymerase. Here, we examine the effects of DinB2 and DinB3 overexpression in mycobacterial cells. We demonstrate that DinB2 can drive diverse substitution mutations conferring antibiotic resistance. DinB2 induces frameshift mutations in homopolymeric sequences, both in vitro and in vivo. DinB2 switches from less to more mutagenic in the presence of manganese in vitro. This study indicates that DinB2 may contribute to mycobacterial mutagenesis and antibiotic resistance acquisition in combination with DinB1 and DnaE2.

**\*For correspondence:**
shumans@mskcc.org (SS);
glickmam@mskcc.org (MSG)

†These authors contributed equally to this work

## Editor's evaluation

This important study uses a combination of compelling biochemical and genetic approaches to identify a highly mutagenic mycobacterial DNA polymerase, which drives a wide spectrum of mutations when overexpressed. The findings advance the understanding of mutagenesis in mycobacteria. The work will be of interest to bacteriologists working on mutation mechanisms and the emergence of drug resistance.

## Introduction

One of the primary mediators of chromosomal mutagenesis is the error-prone DNA damage tolerance pathway termed translesion synthesis (TLS) (*Vaisman and Woodgate, 2017*). TLS polymerases transiently replace the replicative DNA polymerase to traverse lesions that block replication. Because of the flexibility of their active site and lack of proofreading activity, these polymerases facilitate survival during DNA damage but are mutagenic. TLS has been extensively studied in *Escherichia coli*, which encodes two DNA damage-inducible polymerases: DinB (Pol IV) and UmuDC (Pol V) (*Fuchs and Fujii, 2013*; *Fujii and Fuchs, 2020*). DinB and UmuDC confer tolerance to multiple forms of DNA damage and mediate mutagenesis by incorporating both substitution and frameshift mutations. Whereas DinBs are ubiquitous in bacteria, the distribution of UmuDC is more restricted. Many bacteria, including

mycobacteria, encode a second copy of the replicative DNA polymerase (Pol III), called DnaE2 (*Cole et al., 1998*; *Erill et al., 2006*), as a translesion polymerase.

The *Mycobacterium* genus, which includes the causative agent of tuberculosis (TB) *Mycobacterium tuberculosis* (Mtb) and several others pathogens, encodes DnaE2 as well as several DinB paralogs (*Cole et al., 1998*; *Timinskas and Venclovas, 2019*). DnaE2 confers UV tolerance and antibiotic resistance in Mtb through its mutagenic activity and plays a role in pathogenicity (*Boshoff et al., 2003*). DnaE2 was initially thought to be the only active TLS polymerase in mycobacteria. The role of DinBs in DNA damage tolerance and mutagenesis was unclear because initial studies of *dinB* deletion strains failed to identify a function of Mtb DinBs (*Kana et al., 2010*). Mycobacterial DinBs exemplify three phylogenetic subfamilies found in many actinobacteria: DinB1/DinX, DinB2/DinP, and DinB3/msDinB3 (*Cole et al., 1998*; *Timinskas and Venclovas, 2019*). In a recent study, we revealed that *Mycobacterium smegmatis* and Mtb DinB1 contribute to alkylation damage tolerance, antibiotic resistance though a characteristic mutagenic spectrum, and chromosome diversification through frameshift mutations in homo-oligonucleotide runs (*Dupuy et al., 2022*). Some of these DinB1 activities, particularly homopolymeric run frameshift mutagenesis, are redundant with DnaE2 during exposure to DNA damage, suggesting that DinB1 and DnaE2, both of which interact with the β-clamp, exert their overlapping activities at the replication fork.

*M. smegmatis* encodes two additional DinBs, DinB2 and DinB3, whereas Mtb encodes DinB2 but lacks DinB3. Biochemical studies of *M. smegmatis* DinB2 and DinB3 showed that both have polymerase activity (*Ordonez et al., 2014*). DinB2 is notable for its capacity to utilize ribonucleotides during templated DNA synthesis, an activity that is attributable to the absence of a polar filter (*Johnson et al., 2019*) and a permissive steric gate (*Ordonez et al., 2014*). Leucine 14 of DinB2 replaces the canonical aromatic amino acid which clashes with the 2'-OH of rNTPs and thereby confers selectivity to DNA polymerases. DinB2 also displays a metal-dependent mutagenic switch in which manganese supports efficient 8-oxoguanine utilization and nucleotide addition opposite 8-oxoguanine in the template (*Ordonez and Shuman, 2014*). These biochemical activities suggest that DinB2 is equipped to mediate a diverse range of mutation types in vivo. However, our prior data (*Dupuy et al., 2022*) indicate that DinB2 is expressed at very low levels in basal conditions and not induced by DNA damage, in contrast to DnaE2, which is induced ~100-fold by UV. Indeed, *Patra et al., 2021* showed that DinB2 expression is actively repressed in *M. smegmatis* by the action of the TetR family repressor protein MSMEG_2294 encoded in an operon with the *dinB2* gene. This explains why we did not detect any effects of deleting the *dinB2* ORF on mutagenesis in vivo (*Dupuy et al., 2022*), because DinB2 is effectively absent under the conditions surveyed. Thus, an alternative approach is needed.

In this study, we used inducible overexpression to gauge whether and how DinB2 (and DinB3) can affect genomic integrity. We show that DinB2 and DinB3 are both promutagenic in vivo, with distinct mutagenic signatures. In addition, we highlight the strong ability of DinB2, but not DinB3, to incorporate frameshift mutations in both short and long homo-oligonucleotide runs, an activity that is enhanced by manganese in vitro. Finally, we find that manganese enhances the growth inhibitory effects of DinB2 overexpression, suggesting that the metal switch operates in vivo.

## Results

### Overexpression of DinB2 causes cell death through its polymerase activity

To examine the function of DinB2 and DinB3 in *M. smegmatis*, we expressed plasmid-borne copies of the *dinB2* or *dinB3* genes, encoding untagged or streptavidin-tagged (ST) versions of *M. smegmatis* DinB2 and DinB3, under the control of an anhydrotetracycline (ATc)-inducible promoter (tet promoter). DinB2 and DinB3 were detected by immunoblotting with anti-ST antibodies after inducer addition. Levels of DinB2 and DinB3 were similar 4 and 24 hr after ATc treatment (*Figure 1—figure supplement 1A*). Compared to the untreated condition, the levels of DinB2 and DinB3 were increased by between 5- and 15-fold, depending on ATc concentration (*Figure 1A*). Overexpression of tagged and untagged versions of DinB2, but not DinB3, caused a growth defect (*Figure 1B* and *Figure 1—figure supplement 1B*) and 10-fold loss of viability (*Figure 1C*) which was proportional to the concentration of inducer (*Figure 1—figure supplement 1B–C*). DinB2 overexpression also triggered the

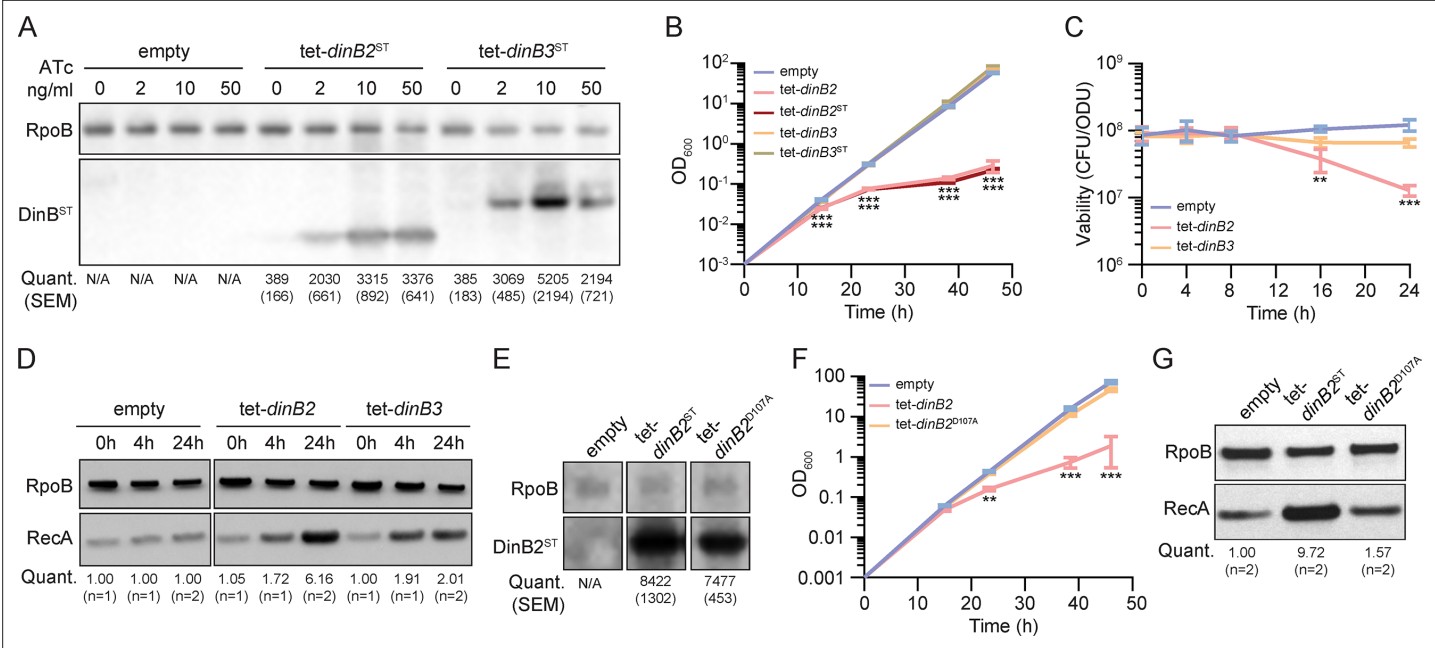

**Figure 1.** Overexpression of DinB2 causes cell death through its polymerase activity. (**A**) Anti-streptavidin/RpoB immunoblots from indicated strains with indicated concentrations of inducer treatment (16 hr of treatment). Average and SEM of RpoB normalized band intensities (n=3, arbitrary units) are given below the image of a representative blot. (**B**) Growth and (**C**) viability of indicated strains in presence of 50 nM anhydrotetracycline (ATc). Note that the OD600 values in (**B**) are calculated values based on continuous dilution growth experiments (see Methods). (**D**) Anti-RecA/RpoB immunoblots from indicated strains with indicated times of inducer treatment (50 nM). Average of normalized band intensities, expressed relative to the empty vector strain, is given below the image of a representative blot. (**E**) Anti-streptavidin/RpoB immunoblots from indicated strains after 16 hr of inducer treatment (50 nM ATc). Average and SEM of normalized band intensities (n=3) are given below the image of a representative blot. (**F**) Growth of indicated strains in presence of 50 nM ATc. (**G**) Anti-RecA/RpoB immunoblots from indicated strains after 24 hr of inducer treatment. Average of normalized band intensities, expressed relatively to the empty vector strain, is given below the image of a representative blot. Empty = empty vector, tet = ATc-inducible promoter, DinB2=*M. smegmatis* DinB2, DinB3=*M. smegmatis* DinB3, ST=streptavidin tag, D107A=catalytically inactive *M. smegmatis* DinB2. Results shown are means (± SEM) of biological triplicates. Stars under the means mark a statistical difference compared to the empty vector reference strain (**, p<0.01; ***, p<0.001).

The online version of this article includes the following source data and figure supplement(s) for figure 1:

**Source data 1.** Uncropped immunoblots *Figure 1A, D, E and G*.

**Figure supplement 1.** DinB2 overexpression induces growth defect in *M. smegmatis*.

**Figure supplement 1—source data 1.** Uncropped immunoblots *Figure 1—figure supplement 1A*.

**Figure supplement 2.** Lethality of DinB2 overexpression in absence of anti-8-oxoguanine systems.

DNA damage response, as measured by RecA induction (*Figure 1D*). Despite no effect on growth, overexpression of DinB3 also induced the DNA damage response (*Figure 1D*).

Our prior work showed that overexpression of DinB1 in *M. smegmatis* also induces cell death, probably by interacting with the replicative machinery and competing with the replicative DNA polymerase (*Dupuy et al., 2022*), a phenotype that was exacerbated by a polymerase dead active site mutation in DinB1. Unlike DinB1, neither DinB2 nor DinB3 have a predicted β clamp binding motif (*Kana et al., 2010*). To investigate the cause of the cell death induced by DinB2 overexpression, we expressed DinB2-D107A, which lacks polymerase activity (*Ordonez et al., 2014*). Whereas DinB2 and DinB2^D107A were expressed at similar levels (*Figure 1E*), DinB2^D107A did not arrest bacterial growth (*Figure 1F* and *Figure 1—figure supplement 1D*) or induce the DNA damage response (*Figure 1G*). These results indicate that, unlike DinB1, the toxic effect of DinB2 is due to its polymerase activity.

A study conducted in *E. coli* revealed that *dinB* overexpression toxicity is due to genomic incorporation and excision of 8-oxoguanines, leading to the formation of DNA double-strand breaks (*Foti et al., 2012*). This is reminiscent of the ability of DinB2 to utilize 8-oxoguanine for DNA synthesis in vitro (*Ordonez and Shuman, 2014*). To test if DinB2 overexpression causes cell death in vivo because of genomic incorporation of 8-oxoguanines, we measured the impact of *mutT*s and *mutY/mutM*s

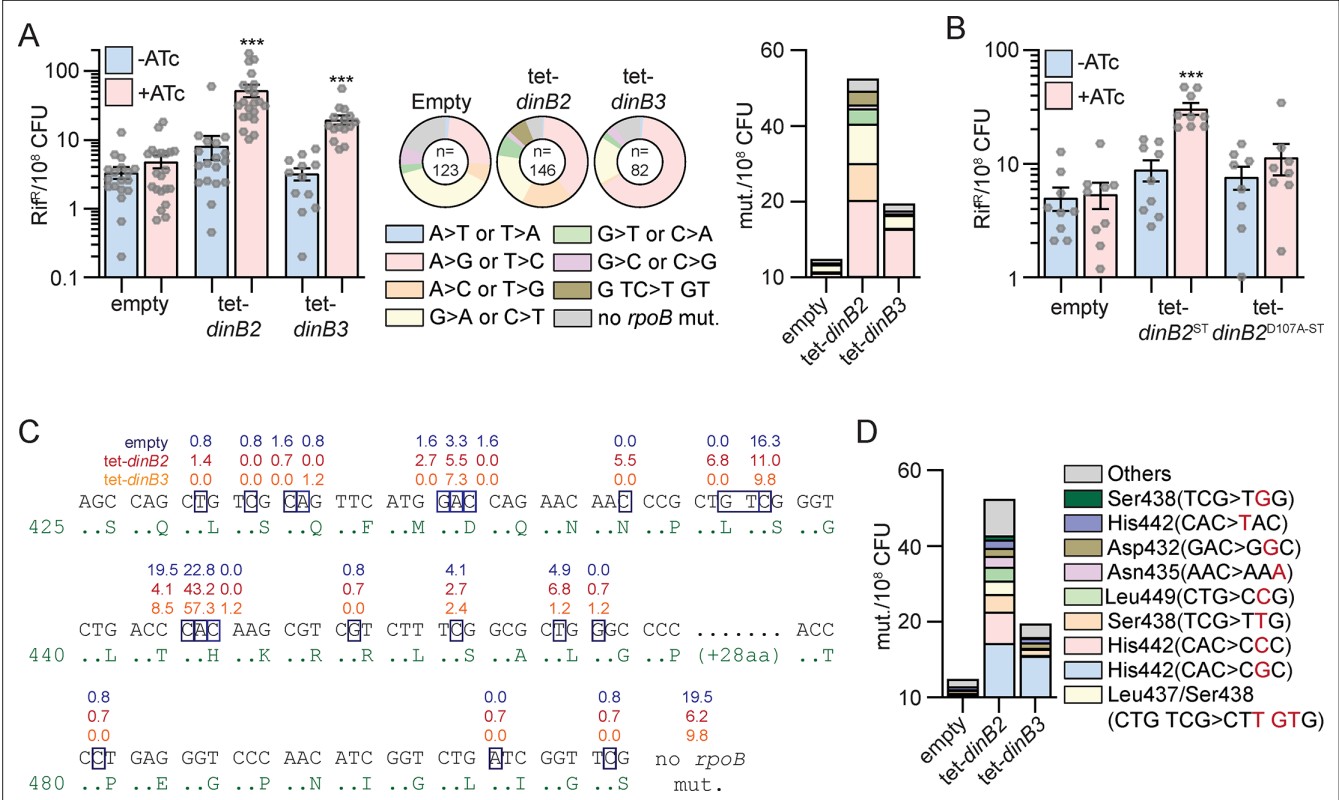

**Figure 2.** DinB2 and DinB3 overexpression confers antibiotic resistance through a distinct mutagenic profile. (**A** and **B**) Rifampicin resistance (rif^R) frequency in indicated strains in absence (blue) or presence (red) of inducer (50 nM anhydrotetracycline [ATc]). Results shown are means (± SEM) of data obtained from biological replicates symbolized by gray dots. Stars above bars mark a statistical difference with the reference (same strain without inducer) (***, p<0.001). Pie charts and bar chart in (**A**) shows the relative and absolute frequencies of nucleotide changes, represented with colors, detected in *rpoB* of indicated strains rif^R in presence of inducer (50 nM ATc). The number of sequenced rif^R is given in the center of each pie chart. (**C**) Location and relative frequency in % of mutated nucleotides in *rpoB* found in empty (blue), tet-*dinB2* (red), or tet-*dinB3* (orange) rif^R. (**D**) Absolute frequency of the main *rpoB* mutations found in indicated strains in presence of 50 nM ATc. Empty = empty vector, tet = ATc-inducible promoter, DinB2=*M. smegmatis* DinB2, DinB3=*M. smegmatis* DinB3, ST = streptavidin tag, D107A=catalytically inactive *M. smegmatis* DinB2.

The online version of this article includes the following figure supplement(s) for figure 2:

**Figure supplement 1.** Mutation frequency in Δ*recA* and Δ*dnaE2* backgrounds after DinB2 and DinB3 overexpression.

deletion on DinB2-dependant lethality. The *mutT* and *mutY/mutM* enzyme systems are involved in degrading free 8-oxoguanine nucleotides and excision of genomic 8-oxoguanines, respectively (*Dupuy et al., 2020*). We did not observe a significant impact of the absence of these anti-8-oxoguanine systems on the *dinB2* overexpression growth defect (*Figure 1—figure supplement 2A and C*) or loss of viability (*Figure 1—figure supplement 2B and D*), indicating that DinB2-dependent lethality is not mainly due to genomic 8-oxoguanine incorporation.

## DinB2 and DinB3 confer antibiotic resistance through a distinct mutagenic profile

Previous studies showed that mycobacterial DnaE2 and DinB1 confer antibiotic resistance by stimulating chromosomal substitution mutations with distinct mutation signatures (*Boshoff et al., 2003*; *Dupuy et al., 2022*). To investigate if DinB2 and DinB3 are similarly mutagenic, we measured the rifampicin resistance (rif^R) frequency in *M. smegmatis* strains overexpressing DinB2 or DinB3. In the absence of inducer, strains carrying the empty vector, tet-*dinB2* or tet-*dinB3* had similar rif^R frequencies. Although 16 hr of inducer treatment had no effect on the rif^R frequency in the control strain, overexpression of DinB2 or DinB3 increased the rif^R frequency by sixfold, compared to –ATc condition (*Figure 2A*). A similar effect was observed with an ST version of DinB2 but not DinB2^D107A-ST(*Figure 2B*), showing that the polymerase activity of DinB2 is necessary to confer antibiotic resistance. DinB2 and

DinB3 overexpression still enhanced rif^R frequency in the ΔrecA and ΔdnaE2 strains (*Figure 2—figure supplement 1*), indicating that the antibiotic resistance conferred by DinB2 is not an indirect consequence of the activation of the DNA damage response or the induction of the DNA damage-inducible error-prone DnaE2 polymerase.

Rifampicin resistance is conferred by substitution mutations in the rifampin resistance determining region (RRDR) of the *rpoB* gene. To define the mutation spectrum stimulated by DinB2 and DinB3, we sequenced the RRDR of rif^R colonies selected after overexpression of these polymerases (*Figure 2A*). In the strain carrying the empty vector, 38% of mutations conferring rif^R were G>A or C>T and 24% were A>G or T>C, with a minority of other mutations, consistent with our prior report (*Dupuy et al., 2022*). DinB3 overexpression enhanced the relative frequency (expressed as %) of A>G or T>C mutations by 2.7-fold and the absolute frequency (expressed as mutants/$10^8$ CFU) by 11-fold with a minimal effect on other mutation types (*Figure 2A*). In contrast, DinB2 overexpression elicited a more diverse spectrum of mutations. Overexpressed DinB2 enhanced the relative frequency of A>C or T>G and G>T or C>A mutations by 2.8- and 2.3-fold, respectively. In addition, a previously undetected mutation type emerged: a trinucleotide mutation (G TC >T GT) spanning two codons and detected in 6.8% of sequenced rif^R colonies. Overall, the absolute frequency of all mutation types was increased at least threefold after DinB2 overexpression, revealing the ability of the polymerase to stimulate diverse mutation types, in contrast to DinB1 (*Dupuy et al., 2022*).

Mapping of the DinB2 or DinB3 stimulated *rpoB* mutations onto the RRDR sequence revealed that 57% of the mutations incorporated by DinB3 were localized in the second nucleotide of the His442 codon vs 23% at this position in the control (*Figure 2C*). The predominant missense change at this codon was CAC>CGC (His>Arg) (*Figure 2D*), the same mutation stimulated by DinB1 (*Dupuy et al., 2022*). The absolute frequency of this mutation was increased 12-fold after DinB3 overexpression (*Figure 2D*). In contrast, DinB2 associated mutations were more widely distributed in the RRDR, particularly at His442 (CAC>CGC, >CCC, or >TAC), Ser438 (TCG>TTG or >TGG), Leu437/Ser438 (CTG TCG>CTT GTG), Leu449 (CTG>CCG), Asn435 (AAC>AAA), and Asp432 (GAC>GGC) (*Figure 2C and D*).

Overall, our results show that DinB2 and DinB3 can mediate rifampicin resistance but with different mutagenic profiles, with DinB2 driving a broader mutation spectrum. In comparison to our prior results with DinB1 and DnaE2, these data indicate that the mutagenic activities of DinB3 and DinB1 are relatively narrow and focused on His442, whereas DinB2 is similar to DnaE2 in its wider mutagenic spectrum.

## DinB2 is highly prone to backward slippage in runs of A and T in vitro

Our previous study revealed that mycobacterial DinB1 mediates –1 and +1 frameshift mutations in runs of homo-oligonucleotides in vivo through a slippage activity of the polymerase (*Dupuy et al., 2022*). To determine whether DinB2 might have similar properties, we measured the ability of DinB2 to perform slippage in vitro. We reacted purified recombinant DinB2 with a 5' $^{32}$P-labeled primer-template DNA in the presence of dTTP or dATP. The DNA substrate consisted of a 13 bp duplex and a 5'-template tail in which the length of the homo-oligonucleotide contains 4, 6, or 8A or T followed by 3C or 3G (*Figure 3A and B*).

In presence of the A4 template and dTTP only, the reaction generated predominant +5 and minority +6 products (*Figure 3A*) that could be the consequence of either: the fill-in of the A4 run followed by the misincorporation of one or two Ts opposite C; or a backward slippage reaction catalyzed by DinB2 incorporating one or two extra Ts in the 4A template tract. DinB2's slippage activity was more evident during synthesis on the A6 primer-template, generating a ladder of products elongated by 8–17 nucleotides. Progression to the A8 template generated a longer array of slippage products extended by 11–23 nucleotides. Similar results were obtained with template runs of 6 or 8 Ts (*Figure 3B*).

The finding that DinB2 is capable of iterative slippage synthesis on a homo-oligomeric tract when the only dNTP available is that templated by the homo-oligomer does not reflect the situation in vivo where DinB2 will have access to the next correctly templated dNTP. To query whether provision of the next templated nucleotide in vitro suppresses slippage, we included a dideoxy NTP (ddNTP): either ddGTP templated by the run of three C nucleotides following the A4, A6, and A8 tracts or ddCTP templated by the run of three G nucleotides flanking the T4, T6, and T8 tracts. ddNTPs are

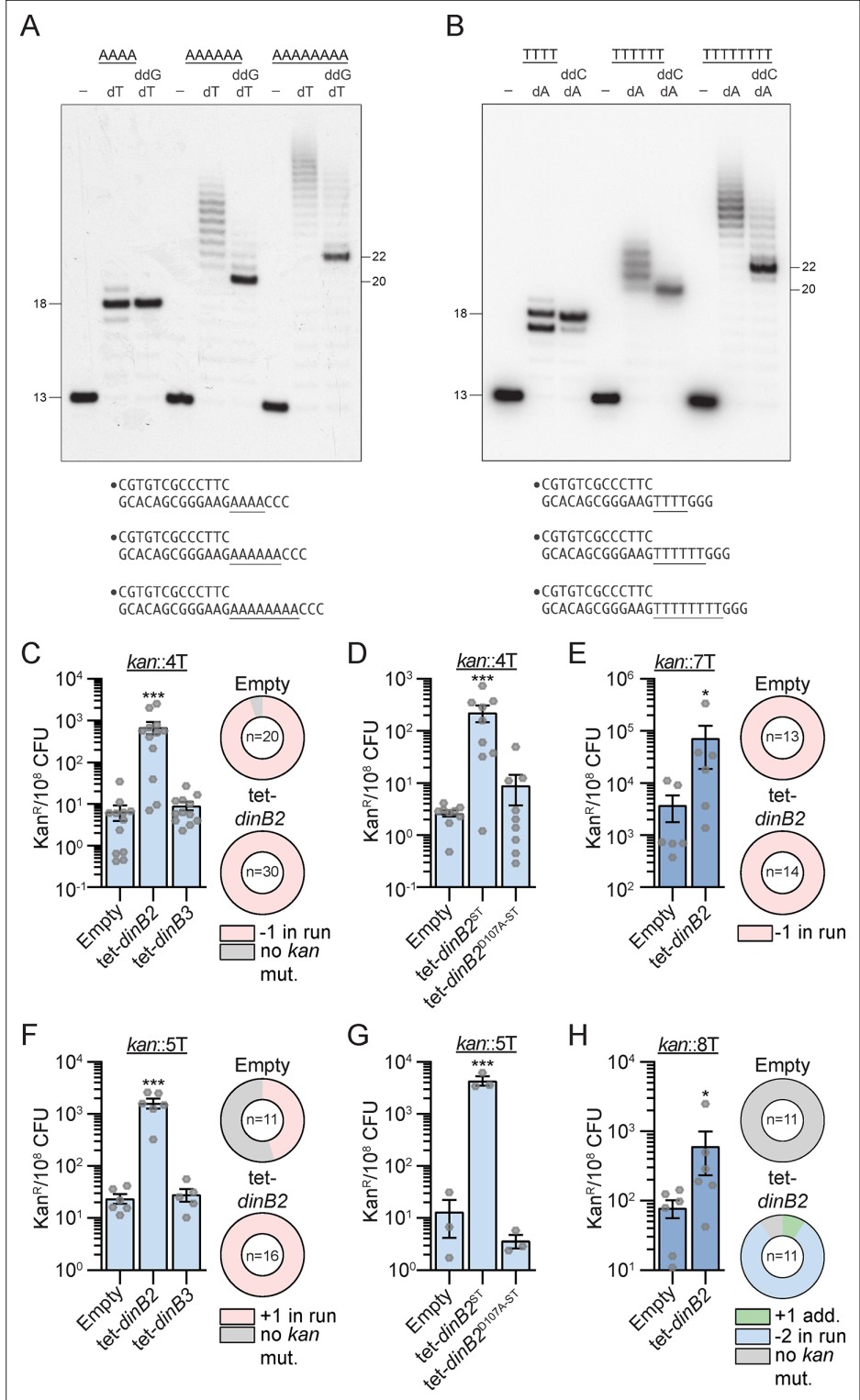

**Figure 3.** DinB2 efficiently promotes –1 and +1 frameshifts in short and long runs of A and T. (**A** and **B**) Reaction mixtures containing 10 mM Tris-HCl, pH 7.5, 5 mM MnCl$_2$, 1 pmol 5' $^{32}$P-labeled primer-template DNAs with indicated runs in the template strand (depicted below, and included as indicated above the lanes), 125 µM dTTP and ddGTP as specified, and 10 pmol DinB2 were incubated at 37°C for 15 min. The reaction products were analyzed by urea-PAGE and visualized by autoradiography. DinB2 was omitted from reactions in lanes –. (**C–H**) kan$^R$ frequencies in the indicated strains carrying the indicated mutation reporters in presence of inducer (50 nM ATc).

*Figure 3 continued on next page*

*Figure 3 continued*

Results shown are means (± SEM) of data obtained from biological replicates symbolized by gray dots. Stars above the bars mark a statistical difference with the reference strain (empty) (*, p<0.05; ***, p<0.001). Relative (pie chart) and absolute (bar chart) frequencies of nucleotide changes detected in *kan* of kan$^R$ cells represented with colors: pink=−1 or +1 frameshift in the homo-oligonucleotide run, blue=-2 frameshift in the run, green=+1 frameshift localized outside of the run and gray=no detected mutation. The number of sequenced kan$^R$ colonies is given in the center of each pie chart. Empty=empty vector, tet = Atc-inducible promoter, DinB2=*M. smegmatis* DinB2, DinB3=*M. smegmatis* DinB3, ST=streptavidin tag, D107A=catalytically inactive *M. smegmatis* DinB2.

The online version of this article includes the following source data and figure supplement(s) for figure 3:

**Source data 1.** Original autoradiograms (*Figure 3A and B*).

**Figure supplement 1.** Frameshift mutagenesis in diverse runs after DinB2 and DinB3 overexpression.

employed to force termination upon incorporation of the first templated nucleotide following the homo-oligomeric tract.

Inclusion of a templated ddGTP with dTTP generated a single +5 extension product on the A4 template, as expected for error-free addition of four dT nucleotides and a single terminal ddG (*Figure 3A*). Similarly, DinB2 reaction with the T4 template in the presence of ddCTP and dATP yielded a predominant +5 extension product reflecting 4 cycles of templated dA incorporation and a single terminal ddC addition (*Figure 3B*). In both cases, the +6 extension products seen with dTTP or dATP only (putative slippage) were suppressed by inclusion of a templated ddNTP.

The evidence for slippage was fortified by the effects of ddGTP on DinB2 activity on the A6 and A8 templates, where ladders of extensions products longer than +7 or +9 nucleotides (the expected results of error-free templated synthesis) were evident (*Figure 3A*). Up to 10 extra dTMP additions were detected on the A8 template even when the next templated nucleotide was available. Similar results applied to the T8 template, whereby ddCTP shortened but did not eliminate the slippage ladder seen with dATP alone (*Figure 3B*). Up to eight extra dAMP additions were observed on the T8 template in the presence of ddCTP. These experiments reveal that DinB2 is prone to backward slippage in A and T runs, a property that is exacerbated in longer homo-oligonucleotide tracts. The heterogeneous size distribution of the slippage ladder is consistent with either of two scenarios: (1) multiple slippage cycles in which the primer 3′-OH end realigns backward on the template by a single nucleotide; and (2) one or several cycles of backward realignment of the primer 3′-OH on the template by more than one nucleotide (the upper limit being the length of the template homo-oligomeric tract) followed by fill-in to the end of the homo-oligomeric tract. It is noteworthy that reaction of DinB2 with the T4, T6, and T8 templates in the presence of dATP and ddCTP also generated a minor elongation product that was 1-nucleotide shorter than the predominant error-free ddC-terminated species, suggestive of a single cycle of forward slippage on the T runs prior to terminal ddC incorporation.

## DinB2 promotes −1 and +1 frameshifts in short and long runs of A and T in vivo

The slippage activity of DinB2 in homo-oligonucleotide runs detected in vitro suggests that the polymerase may incorporate FS mutations in vivo. To test this possibility, we used a reporter tool developed in our previous study (*Dupuy et al., 2022*): a chromosomal integrated *kan* gene conferring the resistance to kanamycin inactivated by an out-of-frame homo-oligonucleotide run immediately downstream of the start codon. FS in the run can restore a functional reading frame and the mutation frequency can be quantified on kanamycin agar. To determine the effect of sequence specificity as well as run size, we used various reporters carrying different runs: 4T (*kan*::4T), 4A (*kan*::4A), 5T (*kan*::5T), 5A (*kan*::5A), 7T (*kan*::7T), 7A (*kan*::7A), 8T (*kan*::8T) and 8A (*kan*::8A).

Using the 4T reporter, we detected 6.6 kanamycin-resistant colonies (kan$^R$) per $10^8$ CFU in the control strain, carrying the empty vector (*Figure 3C*). A majority of the sequenced kan$^R$ had a −1 FS mutation in the run of Ts. The overexpression of DinB2 increased the kan$^R$ frequency more than 100-fold, compared to the control strain, but DinB3 had no effect. All kan$^R$ obtained after DinB2 overexpression had −1 FS localized in the *kan* 4T run, revealing that the polymerase strongly promotes this kind of mutation. The increase of the mutation frequency was reduced by DinB2 active site mutation D107A (*Figure 3D*), indicating that DinB2 directly incorporates −1 FS through its polymerase activity. The −1 FS mutation frequency observed with the 7T reporter in the control strain was much higher than

with the 4T reporter (6.6 vs 3796 kan$^R$/10$^8$ CFU), with 100% of the sequenced kan$^R$ events containing a −1 FS in the run (***Figure 3E***), revealing that the size of the run strongly impacts the spontaneous FS mutation frequency in mycobacteria. Even with this level of background, overexpression of DinB2 enhanced the −1 FS mutation frequency in the 7T run by 19-fold. A similar pattern was observed with runs of 4A and 7A (***Figure 3—figure supplement 1A and B***).

With 5T and 5A reporters, designed to detect +1 FS, the strain carrying the empty vector had 7 and 24 kan$^R$/10$^8$ CFU, respectively, with 45% (5T) and 90% (5A) of sequenced kan$^R$ isolates having +1 FS mutations in the run (***Figure 3F*** and ***Figure 3—figure supplement 1C***). Whereas DinB3 overexpression had no effect on the kan$^R$ frequency, DinB2 overexpression increased the +1 FS frequency by 100-fold in both runs (***Figure 3F*** and ***Figure 3—figure supplement 1C***) and this effect was dependent on its polymerase activity (***Figure 3G***). An increase of the run size (5 vs 8) enhanced the spontaneous +1 FS detected in the run of A by 1000-fold and these events were still induced by DinB2 overexpression (***Figure 3—figure supplement 1D***). In contrast, the kan$^R$ frequency in the strain carrying the empty vector and the 8T reporter was similar to the frequency observed with the 5T reporter, but in contrast to the 5T, the sequenced kan$^R$ had no *kan* mutations (***Figure 3H***), potentially because the encoded three amino acid insertion impairs the function of the Aph protein or expression of the kan$^R$ gene. Despite this finding, we nevertheless observed an eightfold increase of the kan$^R$ frequency after DinB2 overexpression which was due to an accumulation of −2 FS mutations in the 8T run, revealing that DinB2 can also stimulate −2 FS.

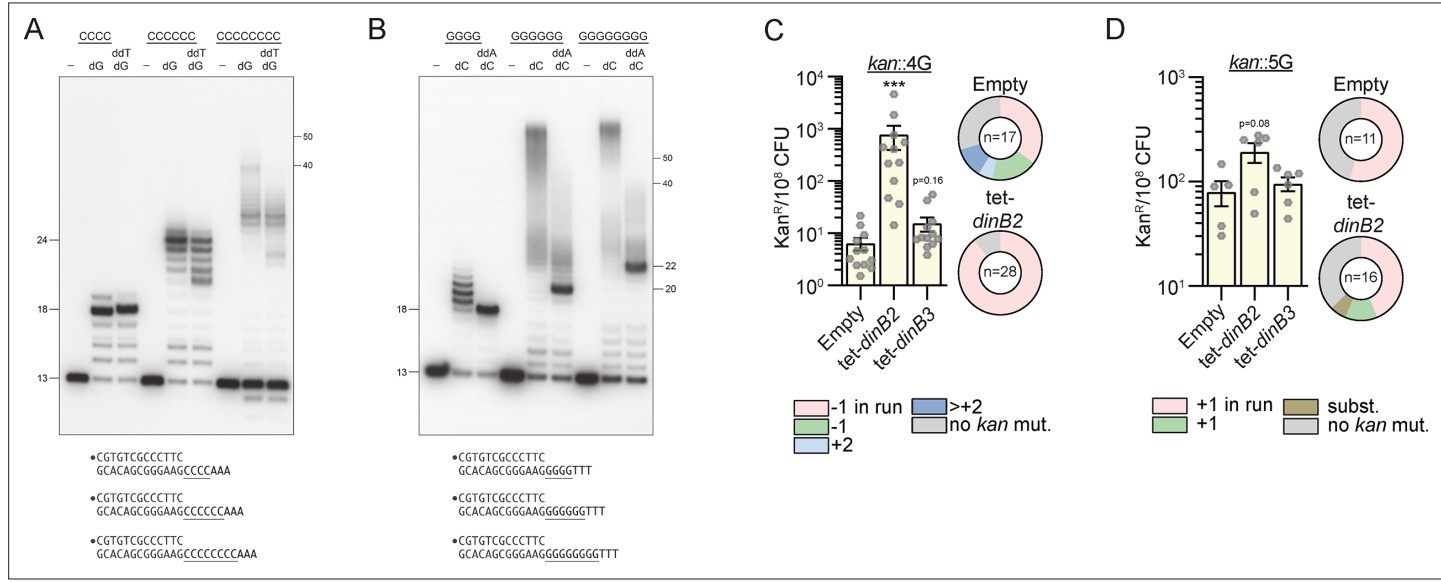

**Figure 4.** DinB2 slippage activity is enhanced on C and G homo-oligonucleotide templates. (**A** and **B**) Reaction mixtures containing 10 mM Tris-HCl, pH 7.5, 5 mM MnCl$_2$, 1 pmol 5' $^{32}$P-labeled primer-template DNAs with indicated runs in the template strand (depicted below, and included as indicated above the lanes), 125 µM dGTP and ddTTP or dCTP and ddATP as specified, and 10 pmol DinB2 were incubated at 37°C for 15 min. DinB2 was omitted from reactions in lanes −. The reaction products were analyzed by urea-PAGE and visualized by autoradiography. The positions of the 13-mer primer strand and 5' $^{32}$P-labeled 40-mer and 50-mer oligonucleotide size markers analyzed in parallel are indicated on the right. (**C–D**) kan$^R$ frequencies in the indicated strains carrying indicated mutation reporters in presence of inducer (50 nM anhydrotetracyclin [ATc]). Results shown are means (± SEM) of data obtained from biological replicates symbolized by gray dots. Stars above the means mark a statistical difference with the reference strain (empty) (***, p<0.001). Relative (pie chart) and absolute (bar chart) frequencies of nucleotide changes detected in *kan* of kan$^R$ cells represented with colors: pink=−1 or +1 frameshift (FS) in the homo-oligonucleotide run, green=−1 or +1 FS localized outside of the run, light blue=+2 FS localized outside of the run, dark blue=>+2 insertion, brown=bases substitution mutation and gray=no detected mutation. The number of sequenced kan$^R$ colonies is given in the center of each pie chart. Empty=empty vector, tet=Atc-inducible promoter, DinB2=*M. smegmatis* DinB2, DinB3=*M. smegmatis* DinB3.

The online version of this article includes the following source data and figure supplement(s) for figure 4:

**Source data 1.** Original autoradiograms (***Figure 4A and B***).

**Figure supplement 1.** Pol1 is not prone to slippage.

**Figure supplement 1—source data 1.** Original autoradiograms (***Figure 4—figure supplement 1A–D***).

**Figure supplement 2.** DinB2 does not incorporate long slippage products in vivo.

Together, these results reveal that the size of homo-oligonucleotide runs strongly impacts the frequency of spontaneous FS mutations incorporated in A and T runs of the mycobacterial chromosome and that DinB2 can catalyze +1, –1, and –2 FS mutagenesis in these low complexity regions.

## DinB2 slippage activity is enhanced on C and G homo-oligonucleotide templates

We proceeded to assay DinB2 with a series of primer-templates containing runs of G or C in vitro. Similar to our findings with 4A and 4T runs, DinB2 synthesis over the 4C run generated +5 and+6 products (*Figure 4A*). However, the polymerase was more prone to slip on the 4G template, generating a cluster of labeled primers extended by 6–9 cycles of dCMP addition (*Figure 4B*). Slippage on G4 was suppressed completely by inclusion of ddATP, which converted the ladder seen with dCTP alone into a single +5 extension product (*Figure 4B*). With the C4 template, inclusion of ddTTP altered the electrophoretic mobility of the error-free ddT-terminated +5 extension product vis-à-vis the +5 dG extension product arising via a single cycle of slippage (*Figure 4A*). A minor +6 extension product in the presence of ddTTP is evidence of residual +1 slippage on the C4 run in the presence of the next templated nucleotide.

Increasing the template G-run or C-run to 6 or 8 nucleotides strongly enhanced the slippage activity of DinB2, generating +12 to >50 products. Although the very longest slippage products were suppressed or diminished by inclusion of ddATP or ddTTP, DinB2 continued to synthesize long products by iterative slippage that contained tracts of 12 to ~30 dCMPs or dGMPs (*Figure 4A and B*).

The finding that DinB2 is more slippery during DNA synthesis across a template G run than on A runs or T runs of equivalent length vitiates the simple hypothesis that recession and realignment of the primer terminus on the template homo-oligonucleotide run is dictated by the thermodynamics of base pairing. If this were the case, we would expect the more weakly paired primer A-tract/template T-tract and primer T-tract/template A-tract configurations in *Figure 3* to be more slippery than the more stably base-paired primer C-tract/template G-tract situation in *Figure 4*. That the opposite scenario applies suggests that DinB2 itself is uniquely geared to slip on G- or C-runs.

These results show that DinB2 is more prone to slippage in vitro than DinB1, that was assessed using a similar assay in our previous study (*Dupuy et al., 2022*). We also found that the polymerase domain of *M. smegmatis* Pol1 (*Ghosh et al., 2020*), homologous to the Klenow polymerase domain of *E. coli* Pol1, was weakly prone to slippage compared to DinB2 (*Figure 4—figure supplement 1*).

## DinB2 slippage in G or C runs is size restricted in vivo

By using the in vivo kan reporter, we investigated the ability of DinB2 and DinB3 to incorporate FS in runs of G. The control strain (empty vector) carrying the 4G reporter had 21.7 kan[R] per $10^8$ CFU and 35% of sequenced kan[R] isolates had a –1 FS in the 4G run (*Figure 4C*). The control strain which carries the 5G reporter showed 79.5 kan[R] per $10^8$ CFU and +1 FS in the run were found in 67% of sequenced kan[R] isolates (*Figure 4D*). Overexpression of *dinB3* did not significantly affect –1 FS or +1 FS mutations localized to the 4G or 5G runs. However, *dinB2* overexpression enhanced –1 FS frequency in the 4G run by 100-fold and a non-statistically significant increase of +1 FS frequency was observed in the 5G run.

Results in *Figure 4A and B* show that DinB2 mostly catalyzes the synthesis of long slippage products, up to 60 nucleotides. These longer products, if they exist in vivo, would not be detected by our *kan* reporter because they will inactivate the Aph enzyme produced by the kan resistance cassette. To investigate if long slippage products are incorporated by DinB2 in vivo, we integrated a *sacB* gene carrying a run of 6C (*sacB*::6C) or 9C (*sacB*::9C) upstream of its start codon into the *M. smegmatis* genome. A functional *sacB* is lethal in presence of sucrose. Selection for sucrose resistance (suc[R]) would reveal all mutagenic events inactivating *sacB*, including putative long slippage events. In the strain carrying the empty vector and the *sacB*::6C reporter, suc[R] colonies were detected at a frequency of 1/10,000 (*Figure 4—figure supplement 2A*). The majority of the suc[R] clones did not have mutation in the 300 first bp of *sacB*, but 15% had a +1 FS in the 6C run and 15% had a substitution generating a stop codon. DinB2 overexpression increased suc[R] frequency by 10-fold and half of the sequenced colonies had +1 FS localized in the 6C run, 7% had a –1 FS in the run, and 7% a substitution confirming that DinB2 promotes both –1 and +1 FS in homo-oligonucleotide runs. However, the incorporation of long slippage products was not detected. In the control strain carrying the *sacB*::9C reporter, 3% of

bacteria were suc$^R$. Among them, 78% had a –1 FS in the run and 22% had a +1 FS in the run (**Figure 4—figure supplement 2B**). Overexpression of DinB2 did not increase the suc$^R$ frequency, which could be due to the high level of background, but the proportion of +1 FS in the run detected among suc$^R$ was enhanced by 2.5-fold. These results indicate that DinB2 does not incorporate long slippage products in vivo in these experimental conditions. We cannot exclude the possibility that the synthesis of long slippage tracks by DinB2, leading to large DNA loops, is lethal for the cell. These results also reveal the very high spontaneous FS frequency (>1%) in runs of nine homo-oligonucleotides.

## DinB2 is less prone to slippage in RNA polymerase mode

DinB2 is the founder of a clade of Y-family DNA polymerase that is naturally adept at incorporating ribonucleotides, by virtue of a leucine in lieu of a canonical aromatic steric gate (**Ordonez et al., 2014**). Incorporation of rNTPs in DNA can lead to genome instability, replication fork blockage, and an increase of the mutation rate including FS (**Kellner and Luke, 2020**). To investigate if the DinB2 phenotypes reported in this work are due to its ribonucleotide utilizing activity, we constructed a restrictive steric gate mutant of DinB2 (L14F) which restores dNTP selectivity (**Ordonez et al., 2014**). The two proteins were expressed at the same level after inducer addition (**Figure 5A**). Overexpression of DinB2$^{L14F}$ resulted in a more substantial growth delay (**Figure 5B**) and enhanced cell death (**Figure 5C**) than overexpression of the WT protein. Induction of the DNA damage response (**Figure 5D**), substitution mutations (**Figure 5E**), or FS mutations (**Figure 5F and G**) were similar after overexpression of DinB2 or DinB2$^{L14F}$. These results show that the mutagenesis phenotypes that accompany DinB2 overexpression are not due to ribonucleotide utilization by DinB2. These data also suggest that FS are directly incorporated by DinB2 and are not the secondary consequence of ribonucleotide excision repair (**Schroeder et al., 2017**).

DinB2 can incorporate at least 16 consecutive ribonucleotides during primer extension on a DNA template (**Ordonez et al., 2014**). We queried whether DinB2 was prone to slippage on homo-oligonucleotide run templates when acting in RNA polymerase mode. DinB2 catalyzed mostly 4, 6, or 8 cycles of rUMP addition on the A4, A6, or A8 template (**Figure 5H**). The reaction on the A4 run revealed a very little extension to +5 that contrasts with the predominance of the +5 addition product generated by DinB2 in DNA polymerase mode (**Figure 3A**). Moreover, unlike the analogous DNA polymerase reaction with dTTP (**Figure 3A**), in the presence of rUMP we detected no slippage products longer than +8 and +10 on the A6 or A8 templates, respectively (**Figure 5H**). When ddGTP was added along with rUTP, DinB2 synthesized the expected +5, +7, and +9 ddG-terminated species (**Figure 5H**). Similar results were obtained when DinB2 was reacted with the G4, G6, and G8 primer-templates in the presence of rCTP (**Figure 5I**). This experiment reveals that DinB2 is much less slippery when utilizing rNTPs.

## Metal-dependent switch in DinB2 slippage

Previous studies showed that DinB2 catalyzes a broader spectrum of deoxynucleotide misincorporations with manganese than magnesium (**Ordonez and Shuman, 2014**). Metal mixing experiments revealed that low ratios of manganese to magnesium sufficed to switch DinB2 to its more mutagenic mode (**Ordonez and Shuman, 2014**). This raises the question of how DinB2 behaves with respect to slippage in presence of manganese, magnesium, or both divalent cations. The in vitro experiments presented above were conducted in presence of 5 mM manganese only (**Figures 3A, B, 4A, B, 5H and I**). The experiment in **Figure 6A** using the A6 primer-template entailed mixing 5 mM magnesium with increasing concentrations of manganese. We observed a manganese binary switch in the product distribution, from a no-slip state with magnesium alone (where the major product is a +7 mis-addition) to a slipped state in which DinB2 catalyzed 8 or more cycles of dTMP addition (**Figure 6A**). This transition was evident at 1 mM manganese, that is, a 1:5 ratio of $Mn^{2+}$ to $Mg^{2+}$. The length of the slippage tract, which was 8–10 nucleotides at 1 mM manganese, increased to 8–14 nucleotides at 2 mM manganese, and saturated at 8–17 nucleotides at 3–5 mM manganese. Note that mixing manganese and magnesium did not diminish the residual slippage events on the A6 template in the presence of ddGTP (**Figure 6A**). A similar metal-dependent switch in DinB2 slippage behavior was seen in experiments using the G6 template (**Figure 6B**), where a finer titration highlighted a transition from no-slippage to slippage at 0.5 mM manganese (a 1:10 ratio of $Mn^{2+}$ to $Mg^{2+}$). Magnesium did not impact the relatively high level of slippage on the G6 template that was seen in the presence of ddATP. We

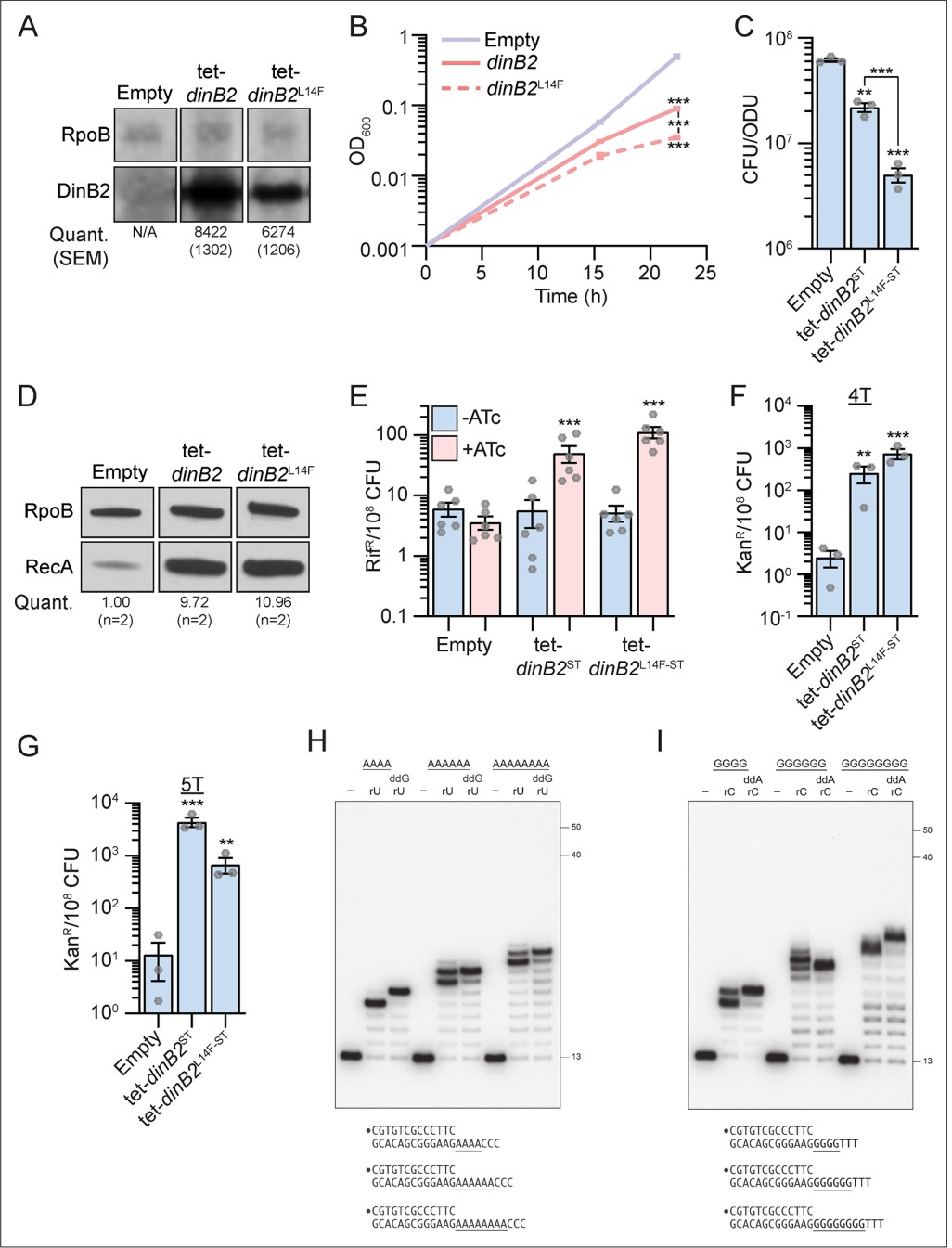

**Figure 5.** DinB2 does not slip in RNA polymerase mode. (**A**) Anti-streptavidin/RpoB immunoblots from indicated strains after 16 hr of inducer treatment (50 nM anhydrotetracycline [ATc]). Average and SEM of RpoB normalized band intensities (n=3) are given below the image of a representative blot. (**B**) Growth of indicated strains in presence of 50 nM ATc. (**C**) Viability of indicated strains after 24 hr of inducer treatment (50 nM ATc). (**D**) Anti-RecA/ RpoB immunoblots from indicated strains after 24 hr of inducer treatment (50 nM ATc). Average of normalized band intensities, expressed relative to the empty condition, is given below the image of a representative blot. (**E**) Rifampicin resistance (rif[R]) frequency in indicated strains in absence (blue) or presence (pink) of 50 nM ATc. (**F** and **G**) kan[R] frequencies in the indicated strains carrying indicated mutation reporters in presence of 50 nM ATc. Results shown are means (± SEM) of data obtained from biological replicates symbolized by gray dots or biological triplicates for (**B**). Stars above the means mark a statistical difference with the reference strain (**B**, **C**, **F**, and **G**: empty or **E**: same strain without inducer). Lines connecting two strains show a statistical difference between them. (**, $p<0.01$; ***, $p<0.001$). (**H** and **I**) Reaction mixtures containing 10 mM Tris-HCl, pH 7.5, 5 mM $MnCl_2$, 1 pmol 5' $^{32}$P-labeled primer-template DNAs with indicated runs in the template strand (depicted below, and included as

*Figure 5 continued on next page*

*Figure 5 continued*

indicated above the lanes), 125 μM rUTP and ddGTP or rCTP and ddATP as specified, and 10 pmol DinB2 were incubated at 37°C for 15 min. DinB2 was omitted from reactions in lanes –. The reaction products were analyzed by urea-PAGE and visualized by autoradiography. The positions of the 13-mer primer strand and 5' $^{32}$P-labeled 40-mer and 50-mer oligonucleotide size markers analyzed in parallel are indicated on the right. Empty=empty vector, tet=Atc-inducible promoter, DinB2=*M. smegmatis* DinB2, ST=streptavidin tag, L14*F*=steric gate mutant of *M. smegmatis* DinB2.

The online version of this article includes the following source data for figure 5:

**Source data 1.** Uncropped immunoblots (*Figure 5A and D*) and original autoradiograms (*Figure 5H and I*).

surmise that DinB2 has a preference for manganese occupancy of at least one of its two metal-binding sites (*Ordonez and Shuman, 2014*) when both magnesium and manganese are present, and that this occupancy suffices to shift DinB2 to a slippage mode.

## Modulation of DinB2 activity in vivo by manganese

To investigate the consequence of the divalent cations nature on the DinB2 activity in vivo, we added increasing concentrations of manganese to Middlebrook 7H9 culture medium which contains 50 mg/L magnesium, with or without induced DinB2 overexpression. Addition of manganese did not impact the DinB2 expression level (*Figure 6C*). Whereas addition of manganese had no effect on the viability of the strain carrying the empty vector, it decreased the viability of the strain overexpressing DinB2 (*Figure 6D*), a phenotype that was reversed by inactivation of DinB2 polymerase activity (*Figure 6E*). We next measured the impact of manganese on DinB2 stimulated frameshifting in vivo. We confirmed that, in absence of manganese, overexpression of DinB2 increased –1 FS or +1 FS localized in runs of 4T or 5T by 100-fold, but addition of manganese did not alter frameshifting (*Figure 6F and G*). Manganese addition also did not impact the DinB2-dependent mutation frequency or type of mutation detected with the *sacB* reporter (*Figure 4—figure supplement 2*). These results indicate that manganese exacerbates the deleterious polymerase activity of DinB2 in vivo but that this effect is likely due to other Mn-dependent activities of DinB2.

## Discussion
### The mutagenic properties of DinB2

DinB2 can execute several types of mutagenesis in vitro, including ribonucleotide insertion in DNA, utilization of 8-oxoguanine, and nucleotide misincorporation (*Ordonez and Shuman, 2014*; *Ordonez et al., 2014*; *Sharma and Nair, 2012*). However, the role of DinB2 in mutagenesis in vivo was unknown and how any DinB2-dependent mutagenic pathway might overlap with the recently described DinB1 pathway had not been examined.

We found that forced expression of DinB3 causes a similar spectrum of base substitutions as DinB1 (*Dupuy et al., 2022*), mainly A>G or T>C mutations. These mutations are focused on single codon in RpoB to produce the His442>Arg mutation. However, DinB2 has a very different mutagenic spectrum, inducing a diversity of mutation types, particularly A>C or T>G and G>T or C>A. This DinB2 effect is dependent on its polymerase activity, whereas we did not express a DinB3 active site mutant. These mutations are the typical signature of 8-oxoguanine presence in DNA (*Dupuy et al., 2020*; *van Loon et al., 2010*) and are consistent with the 8-oxoguanine handling capabilities of DinB2 (*Ordonez and Shuman, 2014*). Moreover, we recently proposed that DinB2, together with other TLS polymerases, contributes to fluoroquinolone bactericidal action by incorporating 8-oxoguanine in *M. smegmatis* genome when the MutT system is inactivated (*Dupuy et al., 2020*). The mutagenesis data presented here further implicates DinB2 in utilizing 8-oxoguanine in vivo under conditions of endogenous or exogenous oxidative stress.

The four most frequent RpoB mutations induced by DinB2 are H442R, H442P, L449P, and S438L. Whereas H442R and S438L mutations are also induced by DinB1, DinB3, or DnaE2 (*Dupuy et al., 2022*), H442P (CAC>CCC) and L449P (CTG>CCG) mutations are more specific to DinB2. In Mtb, the equivalent RpoB mutations are H445P and L452P which comprise 0.1% and 1.2% of rif$^R$ mutations in clinical isolates, respectively (*World Health Organisation, 2021*). Taken together, the mutational spectrum data for DinB1, DnaE2, and DinB2 indicate a division of labor between TLS polymerases

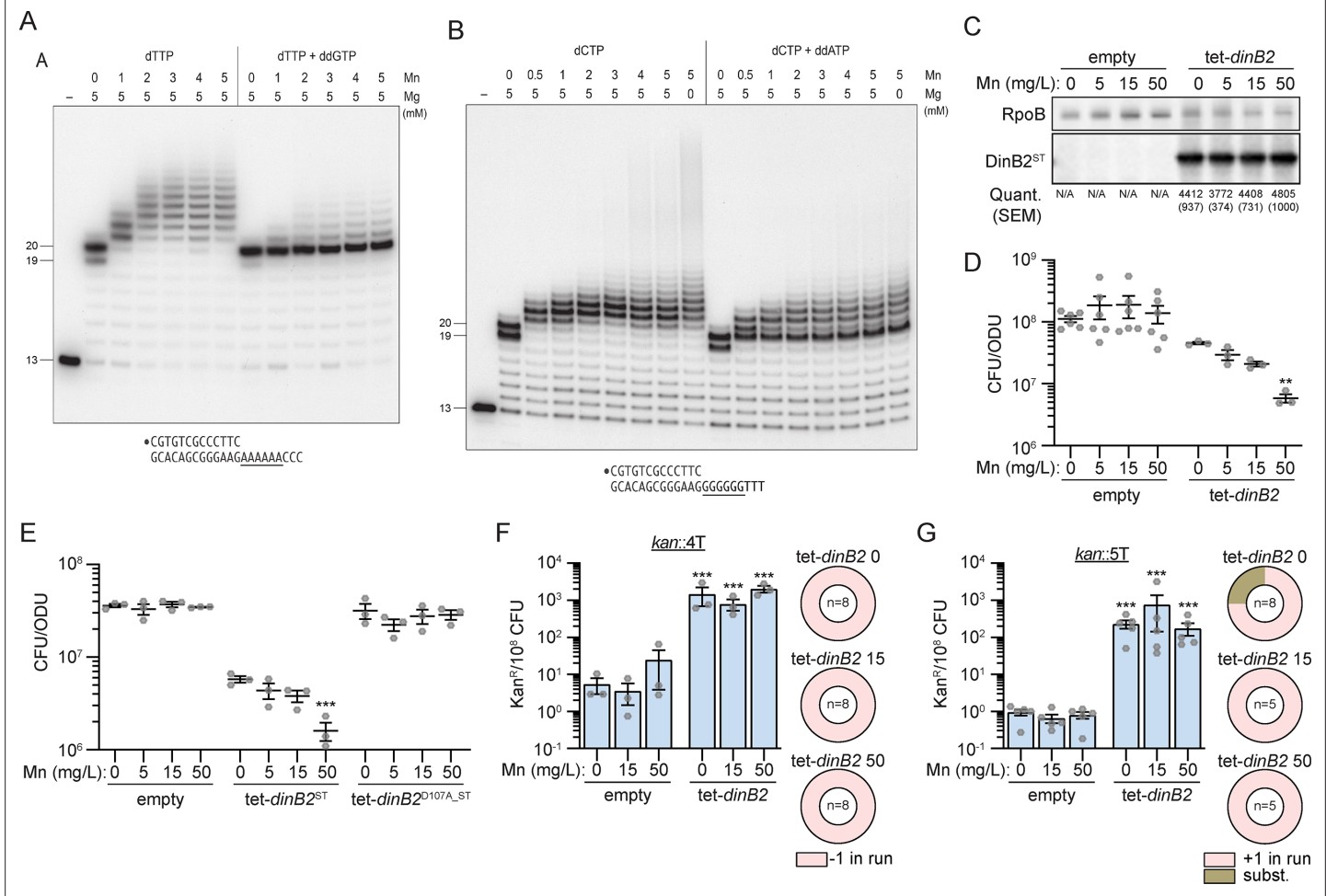

**Figure 6.** Metal-dependent switch in DinB2 activities. (**A** and **B**) Reaction mixtures containing 10 mM Tris-HCl, pH 7.5, 1 pmol 5' $^{32}$P-labeled primer-template DNAs with an A6 or G6 run in the template strand (depicted below), divalent cations and nucleotides (125 μM) as specified above the lanes, and 10 pmol DinB2 were incubated at 37°C for 15 min. DinB2 was omitted from reactions in lanes –. The reaction products were analyzed by urea-PAGE and visualized by autoradiography. (**C**) Anti-streptavidin/RpoB immunoblots from indicated strains cultivated with indicated concentrations of MnCl$_2$ (in mg/L) after 16 hr of inducer treatment (50 nM anhydrotetracyclin [ATc]). Average and SEM of normalized band intensities (n=3) are given below the image of representative blot. (**D** and **E**) Viability of indicated strains after 24 hr of inducer treatment (50 nM ATc) in presence of indicated concentration of MnCl$_2$ (in mg/L). (**F** and **G**) kan$^R$ frequencies in the indicated strains carrying indicated mutation reporters in presence of inducer (50 nM ATc) and indicated concentration of MnCl$_2$ (in mg/L). Results shown are means (± SEM) of data obtained from biological replicates symbolized by gray dots. Relative frequencies of nucleotide changes detected in *kan* of kan$^R$ cells are represented with colors: pink=–1 or +1 frameshift in the homo-oligonucleotide run, brown = substitution mutations. The number of sequenced kan$^R$ colonies is given in the center of each pie chart. Stars above the means mark a statistical difference with the reference strain: same strain untreated with Mn in (**D**) and (**E**), empty with same Mn treatment in (**F**) and (**G**) (**, p<0.01; ***, p<0.001). Empty = empty vector, tet = Atc-inducible promoter, DinB2=*M. smegmatis* DinB2, ST = streptavidin tag, D107A=catalytically inactive *M. smegmatis* DinB2.

The online version of this article includes the following source data for figure 6:

**Source data 1.** Original autoradiograms (*Figure 6A and B*) and uncropped immunoblots (*Figure 6C*).

in the generation of rifampicin resistance in Mtb clinical strains. However, unlike deletion of *dnaE2*, loss of DinBs does not decrease rif$^R$ acquisition in *M. smegmatis* or Mtb in various tested conditions (*Dupuy et al., 2022*; *Kana et al., 2010*), either suggesting redundancy or that a relevant mutagenic stress is missing in culture.

## DinB2 as a catalyst of frameshifting

Because of the high FS frequency observed in mycobacteria (*Gupta and Alland, 2021*; *Springer et al., 2004*) as well as the link of these mutations with antibiotic tolerance in Mtb (*Bellerose et al.,*

*2019*; *Safi et al., 2019*), it is crucial to identify the mechanisms of frameshifting in mycobacteria. Our previous study revealed that DinB1 and DnaE2 promote FS in homo-oligonucleotide runs (*Dupuy et al., 2022*). Here, we showed that DinB2, but not DinB3, also promotes –1 and +1 FS mutagenesis in short (4–5) and long (6–8) homo-oligonucleotide runs in vivo when overexpressed. Compared to DinB1 and DnaE2, overexpression of *dinB2* more efficiently promotes FS mutations in short runs. DinB2 is also more prone to slippage in vitro compared to DinB1 (*Dupuy et al., 2022*) or Pol1 (*Figure 4—figure supplement 1*), generating long backward slippage products (+5 to ~+60) in homo-oligomeric template runs of 6 or 8 nucleotides (*Figures 3A, B, 4A and B*).

By using *dinB1* and *dnaE2* mutants, we previously showed that DinB1 mediates spontaneous –1 FS mutagenesis and DnaE2 is the primary agent of DNA damage-induced –1 FS mutations in *M. smegmatis* (*Dupuy et al., 2022*). However, in that study *dinB2* deletion, alone or in combination with the deletion of other TLS polymerases, did not impact FS mutagenesis in short homonucleotide runs (*Dupuy et al., 2022*). The present work suggests that environmental conditions, including dNTP/rNTP balance or metal ion availability, can influence the mutagenic activity of DinB2 (*Figures 5 and 6*). In addition, recent studies revealed that the *dinB2* expression is under the control of at least two TetR family proteins (Msmeg_6564 and Msmeg_2295), and their deletion and/or overexpression induce *dinB2* expression (*Yang et al., 2012*; *Patra et al., 2021*). Menaquinone, a respiratory lipoquinone, induces *dinB2* expression more than 10-fold, by both inhibiting the Msmeg_2295 repression and acting as a direct inducer (*Barman et al., 2022*). Other experimental conditions will need to be tested to assess the impact of *dinB2* deletion on mutagenesis in mycobacteria, including during infection or after menaquinone treatment, to confirm the phenotypes demonstrated here which are derived from induced expression of DinB2.

## Frameshifts and homo-oligonucleotide runs in mycobacterial genomes

Our study is the first to systematically examine the influence of run length and nucleotide composition on frameshift mutations in mycobacterial homopolymeric sequences. Our data demonstrate that homo-oligonucleotide runs promote –1 and +1 FS mutations and that the length of the run strongly impacts the frequency of FS. Depending on the reporter, we obtained spontaneous FS frequency between $1/10^7$ and $1/10^6$ per CFU in short runs (4–5 nucleotides) and between $1/10^5$ and $1/10^2$ per CFU in long runs (7–9 nucleotides). Our study complements two in silico analyses showing that frameshifts, but not general mutation rates, are higher in mycobacterial genomes than other organisms (*Springer et al., 2004*) and that indels are significantly enriched in homo-oligonucleotide runs of clinical Mtb isolates (*Gupta and Alland, 2021*). A reasonable hypothesis explaining this remarkable frameshift frequency is the absence of a conventional MutS/L mismatch repair in mycobacteria. A study in yeast showed that FS increases 400-fold in a run of 14A relative to a run of 4A in a WT strain which contrasts with the 51,000-fold increase obtained in a mismatch repair mutant (*msh2*), confirming that mismatch repair is a key mechanism for correction of FS in homo-oligonucleotide runs, particularly in long tracks (*Tran et al., 1997*). However, in mycobacteria MutS/L is replaced by an alternative system which depends on NucS and corrects substitution mutations but not FS (*Castañeda-García et al., 2017*; *Castañeda-García et al., 2020*).

Recent data has highlighted the frequency of homopolymeric sequences in mycobacterial genomes and hinted at the functional consequences of sequence variation at these sites. Transient Mtb drug tolerance can be conferred by an FS mutation in a run of C (between 7C and 10C depending on the Mtb strain) of the *glpK* gene, encoding glycerol kinase (*Bellerose et al., 2019*; *Safi et al., 2019*). In addition, recent in silico analyses of genomes of Mtb clinical isolates identified frequent indels in homopolymeric tracts, some of which may have important virulence functions, revealing that reversible gene silencing mediated by FS is not restricted to the *glpK*-dependent antibiotic tolerance pathway (*Gupta and Alland, 2021*; *Vargas et al., 2022*). Based on the level of FS frequency we detected in homo-oligonucleotide runs, we believe that reversible gene silencing through frameshifting in long runs is a prevalent phenomenon in mycobacteria and that DinB1 and DinB2 could contribute to these indels. Further studies are needed to explore the biological functions of reversible gene silencing in mycobacteria, and the roles of DinBs in this process, particularly in genes that contain long homo-oligonucleotide runs.

## A complex network of TLS polymerases in mycobacteria

Mycobacterial DinBs belong to three phylogenetic subfamilies (DinB1, DinB2, and DinB3), distinct from the *E. coli* DinB clade (*Timinskas and Venclovas, 2019*). Timinskas and Venclovas detected 144

bacterial species, almost all actinobacteria, encoding at least one DinB belonging to DinB1, DinB2, or DinB3 family. Among them, 78% encode DinB1, 33% DinB2 and 40% DinB3. DinB1, but not DinB2 or DinB3, interacts with the β clamp and confers tolerance to DNA damage (*Dupuy et al., 2022*; *Kana et al., 2010*), suggesting that DinB1 is the functional ortholog of *E. coli* DinB. Two models are proposed for TLS (*Joseph and Badrinarayanan, 2020*): (1) TLS at the fork during which TLS polymerase switches with the replicative polymerase and assists the replicative machinery in lesion bypass. (2) TLS behind the fork during which the replicative machinery skips past the lesion and continues synthesis downstream, the gap being subsequently filled by the TLS polymerase. *E. coli* DinB seems to mediate both of these pathways (*Joseph and Badrinarayanan, 2020*). A unifying model based on our data is the existence of a division of labor between mycobacterial DinBs, DinB1 mediating TLS at the fork, and DinB2/DinB3 involving in TLS behind the fork. DinB2 and DinB3 could also have other functions, not related to replication, including gap filling during DNA repair. It is intriguing that lower gene dosage of DinB2 in *M. smegmatis* confers susceptibility to dTTP-limiting conditions imposed by expression of a phage ribonucleotide reductase (*Ghosh et al., 2015*). This finding, together with the ability of DinB2 to utilize rNTPs (*Ordonez et al., 2014*), suggests that DinB2 could play an important role in quiescent cells, when dNTPs are limiting.

The role of DinBs in Mtb during host infection is still unknown. Deletion of *dinB1* and *dinB2* does not reduce bacterial survival in macrophages or in mice (*Kana et al., 2010*). However, redundancy with *dnaE2*, which is important for Mtb virulence (*Boshoff et al., 2003*), has been demonstrated for DNA damage tolerance (*Dupuy et al., 2022*) but never explored in the context of infection. Finally, the ability of mycobacterial DinBs, and particularly DinB2, to incorporate FS mutation in homo-oligonucleotide runs, suggests a role for these DNA polymerases in genome evolution through reversible FS in low complexity sequences (*Bellerose et al., 2019*; *Safi et al., 2019*; *Safi et al., 2020*).

## Methods

### Bacterial strains

Bacterial strains used in this work are listed in key resources table. *E. coli* strains were grown in Luria-Bertani medium at 37°C. *M. smegmatis* strains were cultivated at 37°C in Middlebrook 7H9 or 7H10 media (Difco) supplemented with 0.5% glycerol, 0.5% dextrose, and 0.1% Tween 80 (for 7H9 only). Hygromycin was used at 50 µg/mL.

### Plasmids

Plasmids and primers used in this work are listed in key resources table. *kan* plasmids (−1 or +1 frameshift mutation reporters) were constructed as reported in *Dupuy et al., 2022*, using primers listed in key resources table. For *sacB* plasmids (unbiased frameshift mutation reporters), *sacB* was amplified using pAJF067 as a template and primers listed in key resources table and were cloned into pDB60 digested with EcoR1 using recombination-based cloning (In-Fusion, Takara). For *dinB2* and *dinB3* overexpression plasmids constructs, ORFs were amplified using primers listed in key resources table and *M. smegmatis* mc²155 genomic DNA as a template and were cloned into pmsg419 digested with ClaI using In-fusion cloning. Cloning was performed in *E. coli* DH5α and plasmids were introduced in *M. smegmatis* by electroporation.

### Growth and cell viability

Bacteria were grown in absence of inducer (ATc) and diluted to a calculated $OD_{600}$=0.001 in fresh medium supplemented with 50 nM ATc (indicated on the figure when other concentrations were used). $OD_{600}$ was measured for 48 hr. Cultures were diluted in fresh medium supplemented with ATc to a calculated $OD_{600}$=0.001 when $OD_{600}$ reached a value around 1 and measured $OD_{600}$ was corrected by the dilution factor. For viability experiments, ATc was added to exponential growth phase cultures for variable periods of time. Then, $OD_{600}$ was measured and cells were washed in −ATc fresh medium and plated on −ATc agar medium for CFU enumeration. Viability was expressed in CFU number per optical density unit (CFU in 1 mL of a culture at $OD_{600}$=1). For the experiments in *Figure 6*, $MnCl_2$ was added to the +ATc culture, but not to the agar media. For agar medium growth experiments, log-phase cells were cultivated in liquid medium without inducer and serial dilutions were spotted

(5 µL) on agar medium supplemented with 0, 2.5, 5, or 50 nM ATc. Plates were pictured after 72 hr of incubation at 37°C.

## Western blot

Two mL aliquots of a log-phase culture at $OD_{600}$ of 0.4, treated or not with ATc, were used for cell lysate preparation. Lysate preparation and protein separation was conducted as described in *Dupuy et al., 2022*. Blots were blocked and probed in 5% Omniblot milk (RecA detection) or 3% BSA (ST DinBs detection) in PBST (PBS buffer supplemented with 0.1% Tween 20). Proteins on blots were detected using anti-RpoB (Biolegend; 663905; AB_2566583), anti-RecA (Pocono Rabbit Farm & Laboratory, *Wipperman et al., 2018*), or anti-streptavidin (STII GenScript rabbit anti-NWSHPQFEK polyclonal antibody, *Adefisayo et al., 2021*) antibodies incubated at a 1:10,000 dilution for 1 hr and secondary horseradish peroxidase antibodies. Membranes were treated with Amersham ECL western blotting detection reagents (GE Healthcare) according to the manufacturer's instructions Blots were imaged with an iBright FL1000 (Invitrogen). Band intensities were within the linear range of detection and were quantitated with ImageJ in relation to RpoB loading controls.

## Substitution and frameshift mutation frequency

Log-phase bacteria cultured from a single colony were diluted to $OD_{600}$ of 0.004 in fresh medium supplemented or not with 50 nM ATc and/or variable concentration of $MnCl_2$. After 16 hr of culture ($OD_{600}$ ~0.5), cells were concentrated 20-fold by centrifugation and pellet resuspension and 100 µL of a $10^{-6}$ dilution was plated on 7H10 agar whereas 200 µL was plated on 7H10 supplemented with 100 µg/mL rif (substitution mutations), 4 µg/mL Kan (–1 or +1 FS mutations), or 5% sucrose (unbiased FS mutations). Mutation frequency was expressed by the average number of selected colonies per $10^8$ CFU from independent cultures. The number of independent cultures used to measure the mutation frequency of each strain is indicated by the number of gray dots in each graph. The mutation spectrum was determined by amplification and sequencing of the RRDR of the *rpoB* gene of isolated rif$^R$ colonies, the *kan* gene of isolated kan$^R$ colonies, or the *sacB* gene of isolated suc$^R$ colonies. Sequenced rif$^R$, kan$^R$, or suc$^R$ colonies were selected from at least three independent bacterial cultures (between 2 and 8 clones per independent culture were sequenced). The number of sequenced isolated colonies is given in the center of each pie chart. Primers used for amplification and sequencing are listed in key resources table. Mutation spectrum is expressed as relative frequency (percent of mutation types) or absolute frequency (number of each mutation type per $10^8$ CFU obtained by multiplying the rif$^R$ or kan$^R$ frequency by the relative frequency of the mutation type).

## In vitro DNA slippage assay

DinB2 and the C-terminal POL domain of Pol1 were produced in *E. coli* and purified as described previously (*Ordonez et al., 2014*; *Ghosh et al., 2020*). Protein concentrations were determined by using the Bio-Rad dye reagent with bovine serum albumin as the standard. A 5' $^{32}$P-labeled primer DNA strand was prepared by reaction of a 13-mer oligonucleotide with T4 polynucleotide kinase and [γ$^{32}$P]ATP. The labeled DNA was separated from free ATP by electrophoresis through a nondenaturing 18% polyacrylamide gel and then eluted from an excised gel slice. The primer-templates for assays of DNA polymerase were formed by annealing the 5' $^{32}$P-labeled 13-mer pDNA strand to an unlabeled template strand at 1:3 molar ratio. The DNA mixtures were incubated serially at 65°C, 37°C, and 22°C to promote strand annealing. Polymerase reaction mixtures (10 µL) containing 50 mM Tris-HCl, pH 7.5, $MgCl_2$ or $MnCl_2$ as specified, 0.125 mM dNTP, ddNTP, or rNTP as specified, 1 pmol (0.1 µM) $^{32}$P-labeled primer-template DNA, and 10 pmol (1 µM) DinB2 were incubated at 37°C for 15 min. The reactions were quenched by adding 10 µL of 90% formamide, 50 mM EDTA, 0.01% bromophenol blue-xylene cyanol. The samples were heated at 95°C for 5 min and then analyzed by electrophoresis through a 40 cm 18% polyacrylamide gel containing 7.5 M urea in 44.5 mM Tris-borate, pH 8.3, 1 mM EDTA. The products were visualized by autoradiography.

## Statistical analysis

Statistical tests were performed using Prism9 software (GraphPad) on ln-transformed data. Statistical differences in growth and viability experiments were tested using a two-way analysis of variance (ANOVA) and a Bonferroni post-test. Statistical differences in mutagenesis experiments were tested

using a t-test to compare the means of two population groups or a one-way ANOVA and a Bonferroni post-test to compare the means of more than two population groups.

## Acknowledgements

This work is supported by NIH (NIH grant #AI064693) and this research was funded in part through the NIH/NCI Cancer Center Support Grant P30CA008748. P Dupuy was supported in part by a 'Jeune Scientifique' salary award from the French National Institute of Agronomic Science (INRA). We thank all Glickman and Shuman lab members for helpful discussions.

## Additional information

### Competing interests

Michael S Glickman: MG has received consulting fees from Vedanta Biosciences, PRL NYC, and Fimbrion Therapeutics and has equity in Vedanta biosciences. The other authors declare that no competing interests exist.

### Funding

| Funder | Grant reference number | Author |
| --- | --- | --- |
| National Institute of Allergy and Infectious Diseases | AI064693 | Stewart Shuman<br>Michael S Glickman |
| National Cancer Institute | P30CA008748 | Pierre Dupuy<br>Michael S Glickman<br>Stewart Shuman<br>Shreya Ghosh<br>Allison Fay<br>Oyindamola Adefisayo<br>Richa Gupta |
| Institut National de la Recherche Agronomique | Département Caractérisation et Élaboration des Produits Issus de l'Agriculture: Jeune scientifique | Pierre Dupuy |

The funders had no role in study design, data collection and interpretation, or the decision to submit the work for publication.

### Author contributions

Pierre Dupuy, Conceptualization, Formal analysis, Investigation, Methodology, Writing – original draft, Writing – review and editing; Shreya Ghosh, Conceptualization, Formal analysis, Investigation, Methodology, Writing – review and editing; Allison Fay, Investigation; Oyindamola Adefisayo, Conceptualization, Investigation; Richa Gupta, Conceptualization, Investigation, Methodology; Stewart Shuman, Michael S Glickman, Conceptualization, Resources, Supervision, Funding acquisition, Investigation, Writing – original draft, Writing – review and editing

### Author ORCIDs

Pierre Dupuy http://orcid.org/0000-0002-7451-304X
Oyindamola Adefisayo http://orcid.org/0000-0001-9376-148X
Michael S Glickman http://orcid.org/0000-0001-7918-5164

### Decision letter and Author response

Decision letter https://doi.org/10.7554/eLife.83094.sa1
Author response https://doi.org/10.7554/eLife.83094.sa2

## Additional files

### Supplementary files
- MDAR checklist
- Source data 1. Primary data for all non-gel data elements in the figures and figure supplements.

### Data availability
Further information and requests for resources and reagents should be directed to and will be fulfilled by Dr. Michael Glickman (Glickmam@mskcc.org). Plasmids and strains generated in this study will be made available on request. All data generated in this study are presented in the figures and tables and the underlying data supporting all figures is provided as Source data 1.

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

# Appendix 1

## Appendix 1—key resources table

| Reagent type (species) or resource | Designation | Source or reference | Identifiers | Additional information |
|---|---|---|---|---|
| Strain, strain background (*Escherichia coli*) | DH5α | The Glickman lab | | |
| Strain, strain background (*Mycobacterium smegmatis*) | Wild type (mc2155) | *Snapper et al., 1990* | PDS1 | |
| Strain, strain background (*Mycobacterium smegmatis*) | ΔrecA | *Dupuy et al., 2020* | PDS353 | |
| Strain, strain background (*Mycobacterium smegmatis*) | ΔdnaE2 | *Dupuy et al., 2020* | PDS139 | |
| Strain, strain background (*Mycobacterium smegmatis*) | pmsg419 | This work | Mgm4062 | Available from Glickman Lab |
| Strain, strain background (*Mycobacterium smegmatis*) | pRGM47 | This work | mgm4063 | Available from Glickman Lab |
| Strain, strain background (*Mycobacterium smegmatis*) | pRGM48 | This work | mgm4072 | Available from Glickman Lab |
| Strain, strain background (*Mycobacterium smegmatis*) | pRGM49 | This work | mgm4073 | Available from Glickman Lab |
| Strain, strain background (*Mycobacterium smegmatis*) | pRGM50 | This work | mgm4074 | Available from Glickman Lab |
| Strain, strain background (*Mycobacterium smegmatis*) | pDP69 | This work | PDS416 | Available from Glickman Lab |
| Strain, strain background (*Mycobacterium smegmatis*) | pDP70 | This work | PDS417 | Available from Glickman Lab |
| Peptide, recombinant protein | DinB1 | *Ordonez et al., 2014* | | |
| Peptide, recombinant protein | DinB2 | *Ordonez et al., 2014* | | |
| Recombinant DNA reagent | ATc-on system vector (hygR, OriMyc) | Lab Stock | pmsg419 | Available from Glickman Lab |
| Recombinant DNA reagent | Mycob. integr. vector (StrepR, attP(L5)) | Lab Stock | pDB60 | Available from Glickman Lab |
| Recombinant DNA reagent | pmsg419-dinB2 Strep tag | This work | pRGM47 | Cloning primers: dinB2fw-dinB2rev1 (cloning enzyme site: ClaI); available from Glickman Lab |
| Recombinant DNA reagent | pmsg419-dinB2 | This work | pRGM48 | Cloning primers: dinB2fw-dinB2rev2 (cloning enzyme site: ClaI); dinB2=Msmeg_2294; available from Glickman Lab |
| Recombinant DNA reagent | pmsg419-dinB2D107A Strep tag | This work | pRGM49 | Cloning primers: dinB2fw-dinB2catrev +dinB2catfw-dinB2rev1 (cloning enzyme site: ClaI); available from Glickman Lab |

*Appendix 1 Continued on next page*

*Appendix 1 Continued*

| Reagent type (species) or resource | Designation | Source or reference | Identifiers | Additional information |
|---|---|---|---|---|
| Recombinant DNA reagent | pmsg419-dinB2L14F Strep tag | This work | pRGM50 | Cloning primers: dinB2fw-dinB2stericrev +dinB2stericfw-dinB2rev1 (cloning enzyme site: ClaI); available from Glickman Lab |
| Recombinant DNA reagent | pmsg419-dinB3 | This work | pDP69 | Cloning primers: ODP197-ODP298 (cloning enzyme site: ClaI); available from Glickman Lab |
| Recombinant DNA reagent | pmsg419-dinB3 Strep tag | This work | pDP70 | Cloning primers: ODP197-ODP299 (cloning enzyme site: ClaI); available from Glickman Lab |
| Recombinant DNA reagent | pDB60 derivative with kan::3T | *Dupuy et al., 2022* | pDP120 | |
| Recombinant DNA reagent | pDB60 derivative with kan::3C | *Dupuy et al., 2022* | pDP121 | |
| Recombinant DNA reagent | pDB60 derivative with kan::3G | *Dupuy et al., 2022* | pDP122 | |
| Recombinant DNA reagent | pDB60 derivative with kan::3A | *Dupuy et al., 2022* | pDP123 | |
| Recombinant DNA reagent | pDB60 derivative with kan::4T | *Dupuy et al., 2022* | pDP124 | |
| Recombinant DNA reagent | pDB60 derivative with kan::4C | *Dupuy et al., 2022* | pDP125 | |
| Recombinant DNA reagent | pDB60 derivative with kan::4G | *Dupuy et al., 2022* | pDP126 | |
| Recombinant DNA reagent | pDB60 derivative with kan::4A | *Dupuy et al., 2022* | pDP127 | |
| Recombinant DNA reagent | pDB60 derivative with kan::6T | *Dupuy et al., 2022* | pDP128 | |
| Recombinant DNA reagent | pDB60 derivative with kan::6C | *Dupuy et al., 2022* | pDP129 | |
| Recombinant DNA reagent | pDB60 derivative with kan::6G | *Dupuy et al., 2022* | pDP130 | |
| Recombinant DNA reagent | pDB60 derivative with kan::6A | *Dupuy et al., 2022* | pDP131 | |
| Recombinant DNA reagent | pDB60 derivative with kan::7T | This work | pDP132 | Cloning primers: ODP443-ODP445+ODP458-ODP444 (cloning enzyme site: EcoRI); available from Glickman Lab |
| Recombinant DNA reagent | pDB60 derivative with kan::7C | This work | pDP133 | Cloning primers: ODP443-ODP445+ODP459-ODP444 (cloning enzyme site: EcoRI); available from Glickman Lab |
| Recombinant DNA reagent | pDB60 derivative with kan::7G | This work | pDP134 | Cloning primers: ODP443-ODP445+ODP460-ODP444 (cloning enzyme site: EcoRI); available from Glickman Lab |
| Recombinant DNA reagent | pDB60 derivative with kan::7A | This work | pDP135 | Cloning primers: ODP443-ODP445+ODP461-ODP444 (cloning enzyme site: EcoRI); available from Glickman Lab |
| Recombinant DNA reagent | pDB60 derivative with kan::9T | This work | pDP136 | Cloning primers: ODP443-ODP445+ODP462-ODP444 (cloning enzyme site: EcoRI); available from Glickman Lab |
| Recombinant DNA reagent | pDB60 derivative with kan::9C | This work | pDP137 | Cloning primers: ODP443-ODP445+ODP463-ODP444 (cloning enzyme site: EcoRI); available from Glickman Lab |
| Recombinant DNA reagent | pDB60 derivative with kan::9G | This work | pDP138 | Cloning primers: ODP443-ODP445+ODP464-ODP444 (cloning enzyme site: EcoRI); available from Glickman Lab |
| Recombinant DNA reagent | pDB60 derivative with kan::9A | This work | pDP139 | Cloning primers: ODP443-ODP445+ODP465-ODP444 (cloning enzyme site: EcoRI); available from Glickman Lab |
| Recombinant DNA reagent | pDB60 derivative with kan::5T | *Dupuy et al., 2022* | pDP144 | |
| Recombinant DNA reagent | pDB60 derivative with kan::5C | *Dupuy et al., 2022* | pDP145 | |
| Recombinant DNA reagent | pDB60 derivative with kan::5G | *Dupuy et al., 2022* | pDP146 | |
| Recombinant DNA reagent | pDB60 derivative with kan::5A | *Dupuy et al., 2022* | pDP147 | |

*Appendix 1 Continued on next page*

*Appendix 1 Continued*

| Reagent type (species) or resource | Designation | Source or reference | Identifiers | Additional information |
|---|---|---|---|---|
| Recombinant DNA reagent | pDB60 derivative with kan::8T | This work | pDP148 | Cloning primers: ODP443-ODP445+ODP494-ODP444 (cloning enzyme site: EcoRI); available from Glickman Lab |
| Recombinant DNA reagent | pDB60 derivative with kan::8C | This work | pDP149 | Cloning primers: ODP443-ODP445+ODP495-ODP444 (cloning enzyme site: EcoRI); available from Glickman Lab |
| Recombinant DNA reagent | pDB60 derivative with kan::8G | This work | pDP150 | Cloning primers: ODP443-ODP445+ODP496-ODP444 (cloning enzyme site: EcoRI); available from Glickman Lab |
| Recombinant DNA reagent | pDB60 derivative with kan::8A | This work | pDP151 | Cloning primers: ODP443-ODP445+ODP497-ODP444 (cloning enzyme site: EcoRI); available from Glickman Lab |
| Recombinant DNA reagent | pDB60 derivative with sacB::9C | This work | pDP186 | Cloning primers: ODP593-ODP596+ODP597-ODP598 (cloning enzyme site: EcoRI); available from Glickman Lab |
| Recombinant DNA reagent | pDB60 derivative with sacB::6C | This work | pDP194 | Cloning primers: ODP593-ODP614+ODP615-ODP598 (cloning enzyme site: EcoRI); available from Glickman Lab |
| Antibody | Anti-RpoB (mouse monoclonal) | Biolegend | 663905; AB_2566583 | (1:10,000 dilution) |
| Antibody | Anti-RecA (rabbit polyclonal) | *Wipperman et al., 2018* | Anti-RecA | (1:10,000 dilution) |
| Antibody | Anti-streptavidin (rabbit polyclonal) | GenScript | A00626 | STII GenScript rabbit anti-NWSHPQFEK; used at (1:10,000 dilution) |
| Sequence-based reagent | fw dinB2 | IDT | dinB2fw | CAGAAAGGAGGCCATATGACCAAATGGGTGCTC |
| Sequence-based reagent | rev dinB2+streptavidin tag | IDT | dinB2rev1 | AGGTCGACGGTATCGATACTACTTTTCGAACTGCG GGTGGCTCCAGGTGCCTGCAGTGACAG |
| Sequence-based reagent | rev dinB2 | IDT | dinB2rev2 | AGGTCGACGGTATCGATGTGCTCGAGTTAGGTGCCTGCAGTGAC |
| Sequence-based reagent | rev internal dinB2Msm with pol. dead mut. (D107A) | IDT | dinB2catrev | GCCCAGATACGCCTCGGCCCAGCCCCACACCTCCAAC |
| Sequence-based reagent | fw internal dinB2Msm with pol. dead mut. (D107A) | IDT | dinB2catfw | GCCGAGGCGTATCTGGGC |
| Sequence-based reagent | rev internal dinB2Msm with steric gate mut. (L14F) | IDT | dinB2stericrev | GCAACTCCACCGAAGCAAAGAACTGGTCCAGATCGAC |
| Sequence-based reagent | fw internal dinB2Msm with steric gate mut. (L14F) | IDT | dinB2stericfw | TTTGCTTCGGTGGAGTTGC |
| Sequence-based reagent | fw dinB3 | IDT | ODP297 | CAGAAAGGAGGCCATATGTTCGTGTCCGCTGC |
| Sequence-based reagent | rev dinB3 | IDT | ODP298 | AGGTCGACGGTATCGCTAGTCCGGCAGCATGG |
| Sequence-based reagent | rev dinB3+streptavidin tag | IDT | ODP299 | AGGTCGACGGTATCGCTACTTTTCGAACTGCGGGT GGCTCCAGTCCGGCAGCATGGG |
| Sequence-based reagent | fw kan | IDT | ODP443 | TCCAGCTGCAGAATTTCCCAAGGACACTGAGTCC |
| Sequence-based reagent | rev kan | IDT | ODP444 | GATAAGCTTCGAATTTTGCTGACTCATACCAGGC |
| Sequence-based reagent | Internal rev kan | IDT | ODP445 | CATAACACCCCTTGTATTACTG |
| Sequence-based reagent | Internal fw kan (7T addition) | IDT | ODP458 | ACAAGGGGTGTTATGTTTTTTTAGCCATATTCAACGGGAAACG |
| Sequence-based reagent | Internal fw kan (7C addition) | IDT | ODP459 | ACAAGGGGTGTTATGCCCCCCCAGCCATATTCAACGGGAAACG |
| Sequence-based reagent | Internal fw kan (7G addition) | IDT | ODP460 | ACAAGGGGTGTTATGGGGGGGAAGCCATATTCAACGGGAAACG |

*Appendix 1 Continued on next page*

*Appendix 1 Continued*

| Reagent type (species) or resource | Designation | Source or reference | Identifiers | Additional information |
|---|---|---|---|---|
| Sequence-based reagent | Internal fw kan (7A addition) | IDT | ODP461 | ACAAGGGGTGTTATGGAAAAAAAGCCATATTCAACGGGAAACG |
| Sequence-based reagent | Internal fw kan (9T addition) | IDT | ODP462 | ACAAGGGGTGTTATGTTTTTTTTTAGCCATATTCAACGGGAAACG |
| Sequence-based reagent | Internal fw kan (9C addition) | IDT | ODP463 | ACAAGGGGTGTTATGCCCCCCCCCCAGCCATATTCAACGGGAAACG |
| Sequence-based reagent | Internal fw kan (9G addition) | IDT | ODP464 | ACAAGGGGTGTTATGGGGGGGGGGAAGCCATATTCAACGGG AAACG |
| Sequence-based reagent | Internal fw kan (9A addition) | IDT | ODP465 | ACAAGGGGTGTTATGGAAAAAAAAAGCCATATTCAACGGGAAACG |
| Sequence-based reagent | Internal fw kan (8T addition) | IDT | ODP494 | ACAAGGGGTGTTATGTTTTTTTTAGCCATATTCAACGGGAAACG |
| Sequence-based reagent | Internal fw kan (8C addition) | IDT | ODP495 | ACAAGGGGTGTTATGCCCCCCCCCAGCCATATTCAACGGGAAACG |
| Sequence-based reagent | Internal fw kan (8G addition) | IDT | ODP496 | ACAAGGGGTGTTATGGGGGGGGGAAGCCATATTCAACGGGAAACG |
| Sequence-based reagent | Internal fw kan (8A addition) | IDT | ODP497 | ACAAGGGGTGTTATGGAAAAAAAAGCCATATTCAACGGGAAACG |
| Sequence-based reagent | fw sacB | IDT | ODP593 | TCCAGCTGCAGAATTAACCCATCACATATACCTGCCG |
| Sequence-based reagent | Internal rev sacB (9C addition) | IDT | ODP596 | GTTGGGGGGGGGCATCGTTCATGTCTCCTTTTTTATG |
| Sequence-based reagent | Internal fw sacB (9C addition) | IDT | ODP597 | ATGCCCCCCCCCAACATCAAAAAGTTTGCAAAACAAG |
| Sequence-based reagent | rev sacB | IDT | ODP598 | GATAAGCTTCGAATTACTATCAATAAGTTGGAGTCATTACC |
| Sequence-based reagent | Internal rev sacB (6C addition) | IDT | ODP614 | GTTGGGGGGCATCGTTCATGTCTCCTTTTTTATG |
| Sequence-based reagent | Internal fw sacB (6C addition) | IDT | ODP615 | ATGCCCCCCAACATCAAAAAGTTTGCAAAACAAG |
| Sequence-based reagent | fw PCR screening and seq pmsg419 cloning | IDT | ODP236 | CTCCCTATCAGTGATAGATAGGCTCTGG |
| Sequence-based reagent | rev PCR screening and seq pmsg419 cloning | IDT | ODP237 | CATGACCAACTTCGATAACGTTCTCGG |
| Sequence-based reagent | fw PCR screening and seq pDB60 cloning | IDT | ODP474 | TGATTCTGTGGATAACCGTATTACCGCCTTTG |
| Sequence-based reagent | rev PCR screening and seq pDB60 cloning | IDT | ODP475 | AAGGCCCAGTCTTTCGACTGAGC |
| Sequence-based reagent | fw rpoB PCR | IDT | ODP378 | CAAGAAGCTGGGCCTGAACGC |
| Sequence-based reagent | rev rpoB PCR | IDT | ODP379 | GCGGTTGGCGTCGTCGTG |
| Sequence-based reagent | rpoB seq | IDT | ODP380 | GAGCGTGTCGTGCGTGAG |
| Sequence-based reagent | fw kan or sacB PCR | IDT | ODP476 | TGGCCTTTTGCTGGCCTTTTGC |
| Sequence-based reagent | rev kan PCR | IDT | ODP477 | TTCAACAAAGCCGCCGTCCC |
| Sequence-based reagent | kan seq | IDT | ODP479 | ACTGAATCCGGTGAGAATGG |
| Sequence-based reagent | rev sacB PCR | IDT | ODP172 | TTAGACGTAATGCCGTCAATCGTC |

*Appendix 1 Continued*

| Reagent type (species) or resource | Designation | Source or reference | Identifiers | Additional information |
| --- | --- | --- | --- | --- |
| Sequence-based reagent | sacB seq | IDT | ODP474 | TGATTCTGTGGATAACCGTATTACCGCCTTTG |
| Sequence-based reagent | 5' 32P-labeled primer DNA strand | IDT | SG-FS1 | CGTGTCGCCCTTC |
| Sequence-based reagent | Unlabeled template strand (4T) | IDT | SG-FS1 | GGGTTTTGAAGGGCGACACG |
| Sequence-based reagent | Unlabeled template strand (6T) | IDT | SG-FS1 | GGGTTTTTTGAAGGGCGACACG |
| Sequence-based reagent | Unlabeled template strand (8T) | IDT | SG-FS1 | GGGTTTTTTTTGAAGGGCGACACG |
| Sequence-based reagent | Unlabeled template strand (4A) | IDT | SG-FS1 | CCCAAAAGAAGGGCGACAC |
| Sequence-based reagent | Unlabeled template strand (6A) | IDT | SG-FS1 | CCCAAAAAAGAAGGGCGACAC |
| Sequence-based reagent | Unlabeled template strand (8A) | IDT | SG-FS1 | CCCAAAAAAAAGAAGGGCGACAC |

