## [Editor Report]

This important study uses a combination of compelling biochemical and genetic approaches to identify a highly mutagenic mycobacterial DNA polymerase, which drives a wide spectrum of mutations when overexpressed. The findings advance the understanding of mutagenesis in mycobacteria. The work will be of interest to bacteriologists working on mutation mechanisms and the emergence of drug resistance.

---

## [Decision Letter]

**Decision letter after peer review:**

Thank you for submitting your article "Pleiotropic roles for mycobacterial DinB2 in frameshift and substitution mutagenesis" for consideration by *eLife*. Your article has been reviewed by 3 peer reviewers, and the evaluation has been overseen by a Reviewing Editor and Bavesh Kana as the Senior Editor. The following individual involved in the review of your submission has agreed to reveal their identity: Helena Boshoff (Reviewer #2).

Essential revisions:

1. Whilst the results are interesting, overexpression of the different DinBs is an artificial way of looking at their biological roles. Just because they increase mutation frequency when overproduced doesn't mean they act in DNA damage tolerance or mutagenesis when expressed at normal cellular levels from their native genomic loci. While this study is well done, it remains unclear whether DinB2 and/or DinB3 contribute to DNA damage tolerance and mutagenesis in Mycobacteria. Now that the authors have a unique mutation spectrum for DinB2 (when overexpressed), is it possible to search for that spectrum in response to DNA damage, and its disappearance in a strain lacking DinB2 function? This type of result would provide a strong argument that DinB2 is contributing to mutagenesis.

2. The dependence on the catalytic activity of overproduced levels of DinB2 to impede growth is very reminiscent of work from Walker (Foti et al., Science (2012) 336:315-319) concluding that overproduced levels of *E. coli* DinB (Pol IV) kill by incorporating oxidized precursors into nascent DNA, leading to lethal numbers of dsDNA breaks. Have the authors tested whether killing induced by the overproduction of DinB2 can be suppressed by anaerobic growth or similar treatments to limit the production of oxidized DNA precursors?

3. There are several concerns with the Western blots. First, can the authors comment on why panels 1A and 1E look so different? Second, quantitation of the blots would be helpful to gauge the expression levels of the DinBs? For example, was DinB3 expressed at the same level as DinB2, as a function of ATc concentration, and is growth impairment correlated with levels of DinB2 protein and not just ATc? Third, how many times were blots performed? Assuming each represents one of multiple reps their quantitation could indicate the average value +/- error (range or SD). Fourth, by eye, it looks like the DinB2L14F mutant (Figure 5) may be expressed at a slightly lower level than WT DinB2, but it has a more pronounced killing phenotype. Quantitation would support or refute the possibility that the ability to incorporate rNTPs impacts the growth defect caused by DinB2 overproduction. Finally, in Figure 6 was it confirmed that variable metal (Mg, Mn) concentrations failed to influence DinB2 levels by Western blot analysis?

4. The finding of kanR CFUs without any detectable mutations in the kan marker is worrisome and should be better discussed in the text. The same for sacB data in the supplementary material. The explanation given in lines 216-218 does not make sense. Markers 7G and 8G clearly are barely measuring any mutagenesis. The experiments wherein most of the supposed KanR revertants have no Kan mutation should either be removed from the manuscript or better discussed, because it is uncertain what they are measuring, therefore no conclusion can be drawn from them. For the Kan markers, one possible explanation is that translational frameshifts are occurring and allow residual growth of some of the cells. Gene amplification as seen in the lac system of Cairns and Foster in *E. coli* could also promote growth without actual mutations. Is the KanR phenotype of these colonies heritable and stable?

5. Also, spontaneous mutagenesis should have been more precisely measured by using fluctuation analysis of larger sample sizes. In many instances, the results shown are the means of a few cultures with very large differences in mutant frequencies (several hundred-fold – e.g. Figures 4C,D and E, 5C and F, S3). Authors could discuss/explain their choice of statistical analysis and sample sizes.

Other issues that should be addressed:

1. Title: Why are the authors are claiming "pleiotrpic" roles for these polymerases? What are the multiple phenotypes being observed, besides an effect on mutagenesis? The abstract is also not clear about this.

2. Line 44: Error in the citation of Boshoff 2003.

3. Figures 1B and 1F: How OD600 values are measured? Are these very high densities really achievable in mycobacterial cultures (OD600 = 100)?

4. A simple change that would add a lot to the paper. In all gel images showing polymerase activity, an indicator of the molecular size of the relevant products discussed in the text would be of tremendous help to the readers. It is often confusing to follow what is being discussed about the length of slippage tracts without this aid.

5. Lines 169-180: Why do the authors interpret products with an extra 8nt as the result of 8 rounds of slippage? Although this is possible, one slippage event could loop out more than one nucleotide, giving rise to the addition of more than one nucleotide. This idea is repeated later in the text when discussing figure 4.

6. Line 180: "single cycle of forward slippage".

7. Line 249: Klenow polymerase domain.

8. Line 252: "DinB2 slippage in G or C…"

9. Line 259 and 260. The difference observed in the Kan marker 5G is not significant, as indicated in the figure.

10. Lines 333 to 334: "Magnesium did not impact the relatively high level of slippage on the G6 template that was seen in the presence of ddATP." This is not clear, given that the experiment shows a clear effect of Mn ions. Would the point be that increasing concentrations of Mn do not enhance slippage significantly after the transition seen at 0.5 mM?

11. Figures 6E and 6F: indicate which reporters are being used. 4T, 5T?

12. The authors state in Methods that at least 3 cultures were sequenced for each mutagenesis experiment, but how many of the sequences came from each culture?

13. Figure S2 shows results for only DinB2, not DinB2 and DinB3, as stated in the text (lines 104-107).

14. Did the authors examine DinB3 replication on some of the same templates as DinB2, and if so did it also slip? If not, this would further support the conclusion that slippage is the cause of the frameshifts.

15. Line 250: was this slippage or misincorporation?

16. Both DinB2 and DinB3 over-expression result in the induction of RecA. The extent of upregulation of the SOS response is not readily discerned from the RecA immunoblot. Do the authors have any associated transcriptional data to investigate any differences?

Figure S2: the DinB3 over-expression in the dnaE2 and recA deletion mutants is missing but referred to in the text. This is an important control and the panel would be good to include.

17. Manganese supplementation of the culture medium does not increase the frequency of slippage in the induced tet-dinB2 strain and probably does not significantly affect that in the control strain either. A good additional control would have been the mutant lacking dnaE2 and dinB1-3 used in the Dupuy et al. (2022) study.

18. This study demonstrates the role of DinB2-mediated mutagenesis in vivo. It can incorporate ribonucleotides as well as dNTPs. The ratio of these could affect its in vivo activity. One wonders whether the over-expression of DinB2 and the DinB2-L14F (steric mutant) result in altered mutation frequencies when plated from starved cells.

19. Figure 1A vs Figure 5A: why is the DinB immunoblotting so weak in 1A?

20. Line 423: it would be informative to compare homo-oligonucleotide runs in bacteria such as *E. coli* and *B. subtilis* to get a better sense of how common such tracts are between different species.

*Reviewer #1 (Recommendations for the authors):*

This is an interesting study that follows up on a recent important publication from the same group (Dupuy et al., Nature Communications (2022) 13:4493). The M. smegmatis and *M. tuberculosis* DinB proteins (DinB1-DinB3 for Msm; DinB1-DinB2 for Mtb) were previously thought to play no role in DNA damage tolerance and mutagenesis. However, the recent work from Dupuy et al. shows that DinB1 from both organisms plays important roles in DNA damage-induced mutagenesis. By contrast, the authors failed to observe a phenotype for DinB2 and DinB3 mutants. In the current work, the authors have studied the impact of DinB2 and DinB3 overproduction on mutation frequency/signature, concluding that DinB2 plays important roles in both substitution and frameshift mutagenesis and that Mn contributes to the mutator phenotype. Finally, they convincingly demonstrate that frameshifts are the result of DinB2 slipping in homopolymeric nucleotide runs during dNTP and not rNTP incorporation. With the exceptions noted below, this work is well-designed, well-performed, and well-presented. The frameshift results are a strength and represent the most complete and systematic analysis this reviewer has seen. The main concern with the work is that it has studied DinB2 and DinB3 under very artificial conditions that may not represent biologically important (or even relevant) roles for these proteins in M. smegmatis. As such, it remains to be demonstrated whether DinB2 and/or DinB3 contribute to DNA damage tolerance and mutagenesis in vivo when expressed at normal cellular levels from their native genomic loci. This and other concerns the authors should address are discussed below.

1) I appreciate why the authors did what they did, and while I agree their results are interesting, overexpression of the different DinBs is an artificial way of looking at their biological roles. Just because they increase mutation frequency when overproduced doesn't mean they act in DNA damage tolerance or mutagenesis when expressed at normal cellular levels from their native genomic loci. While this study is well done, it remains unclear whether DinB2 and/or DinB3 contribute to DNA damage tolerance and mutagenesis in Mycobacteria. Now that the authors have a unique mutation spectrum for DinB2 (when overexpressed), is it possible to search for that spectrum in response to DNA damage, and its disappearance in a strain lacking DinB2 function? This type of result would provide a strong argument that DinB2 is contributing to mutagenesis.

2) The dependence on catalytic activity of overproduced levels of DinB2 to impede growth is very reminiscent of work from Walker (Foti et al., Science (2012) 336:315-319) concluding that overproduced levels of *E. coli* DinB (Pol IV) kill by incorporating oxidized precursors into nascent DNA, leading to lethal numbers of dsDNA breaks. Have the authors tested whether killing induced by the overproduction of DinB2 can be suppressed by anaerobic growth or similar treatments to limit the production of oxidized DNA precursors?

3) There are several concerns with the Western blots. First, can the authors comment on why panels 1A and 1E look so different? Second, quantitation of the blots would be helpful so the reader knew at what levels the DinBs were expressed. For example, was DinB3 expressed at the same levels as DinB2 as a function of ATc concentration, and is growth impairment correlated with levels of DinB2 protein and not just ATc? Third, how many times were blots performed? Assuming each represents one of multiple reps their quantitation could indicate the average value +/- error (range or SD). Fourth, by eye, it looks like the DinB2L14F mutant (Figure 5) may be expressed at a slightly lower level than WT DinB2, but it has a more pronounced killing phenotype. Quantitation would support or refute the possibility that the ability to incorporate rNTPs impacts the growth defect caused by DinB2 overproduction. Finally, in Figure 6 was it confirmed that variable metal (Mg, Mn) concentrations failed to influence DinB2 levels by Western blot analysis?

4) Several of the gels showing DinB2 replication products are lacking size standards, making it very difficult to follow along with the discussion of the results.

*Reviewer #3 (Recommendations for the authors):*

Title: I do not follow why the authors are claiming "pleiotrpic" roles for these polymerases. What are the multiple phenotypes being observed, besides an effect on mutagenesis? The abstract is also not clear about this.

Line 44: Error in the citation of Boshoff 2003.

Figures 1B and 1F: How OD600 values are measured? Are these very high densities really achievable in mycobacterial cultures (OD600 = 100)?

A simple change that would add a lot to the paper. In all gel images showing polymerase activity, an indicator of the molecular size of the relevant products discussed in the text would be of tremendous help to the readers. It is often confusing to follow what is being discussed about the length of slippage tracts without this aid.

Lines 169-180: I did not understand why authors interpret products with an extra 8nt as the result of 8 rounds of slippage. Although this is possible, one slippage event could loop out more than one nucleotide, giving rise to the addition of more than one nucleotide. This idea is repeated later in the text when discussing figure 4.

Line 180: "single cycle of forward slippage".

Line 249: Klenow polymerase domain.

Line 252: "DinB2 slippage in G or C…"

Line 259 and 260. The difference observed in the Kan marker 5G is not significant, as indicated in the figure.

Lines 333 to 334: "Magnesium did not impact the relatively high level of slippage on the G6 template that was seen in the presence of ddATP.". I did not understand the point raised here, given that the experiment shows a clear effect of Mn ions. Would the point be that increasing concentrations of Mn do not enhance slippage significantly after the transition seen at 0.5 mM?

Figure 6 E and 6F: indicate which reporters are being used. 4T, 5T?

[Editors' note: further revisions were suggested prior to acceptance, as described below.]

Thank you for resubmitting your work entitled "Roles for mycobacterial DinB2 in frameshift and substitution mutagenesis" for further consideration by *eLife*. Your revised article has been evaluated by Bavesh Kana (Senior Editor) and a Reviewing Editor.

The manuscript has been improved but there are some remaining issues that need to be addressed, as outlined below:

Reviewers agreed that the revised manuscript has been substantively improved, there is however one primary concern remaining. This relates to ascribing a function for DinB2 in mutagenesis, particularly the conditions under which such activity manifests. As a first request, please consider revising your discussion, and appending text in the abstract to specifically address this limitation. Refer to the detailed comments from the reviewers to get a better sense of this concern, particularly point 1 from Reviewer 1.

Secondly, the following points need addressing, also through revision of the manuscript to acknowledge any limitations and clarify data presentation:

1. The concern about the variation between western blots in panels 1A vs 1E referred to the variable signal for the RpoD loading control. The weak RpoD signal (in contrast to other RpoD panels) may be outside the linear range of detection, drawing into question the accuracy of its use as a loading control and normalizing standard for the quantitation of RecA. Likewise, in panel A of Figure 1-supplement figure 1, the signal for the DinB2 and DinB3 proteins was very low (in contrast to other DinB2/DinB3 panels) and may not be in the linear range of detection. Can you the authors explain this, specifically with respect to the accuracy of the quantitation of these low signals.

2. Figure 4, supplemental panels 1A-D, and figure 4B would benefit from more labels of molecular sizes for DNA products. For example, in figure 4 the authors state that products were in excess "~+60" in size, but the size markings only go up to 50 bp – it would greatly help the reader to see where the +60 product migrates to.

3. Pg. 6, lines 129-133: DinB1 is regulated by SigH, but can the authors exclude a role for chromosomally-expressed DinB1 in contributing to mutagenesis, similar to the arguments for RecA (SOS) and DnaE2?

4. Pg. 7, lines 158-162: The authors may want to relax/qualify their statement regarding DinB3 as they did not use a DinB3 active site mutation to confirm that mutations observed were in fact the result of DinB3 catalytic activity.

*Reviewer #1 (Recommendations for the authors):*

Recommendations for the authors: While I enjoyed reading this paper, and this revised version is improved, I do have a few comments, some that were raised in the initial review and were not completely addressed and some that are new and are in response to some of the conclusions stated in the revised version.

1) "Essential revision" point #1: a remaining concern is whether levels of the DinB2/B3 proteins examined in this work are higher than when induced from the native genomic loci using, for example, menaquinone to induce dinB2 (I don't believe they discussed the ability of menaquinone to induce dinB2 in their initial draft). I think this point is important because overexpression of a DNA polymerase can confer phenotypes that have nothing to do with its true biological function. For example, the Sweasy lab developed genetic selections and screens for rat DNA polymerase β (Pol β) overexpressed in *E. coli* based on the ability of Pol β to access DNA in a recA polA mutant (e.g., PNAS, 94: 321-1326 and JBC, 274(6): 3851-3858). Thus, while it may be the case, as the authors suggest, that failure to observe a phenotype for the DinB2 mutant may be the result of the conditions studied failing to induce DinB2 (or DinB3) expression, it may also be the case that DinB2/DinB3 fail to confer a mutator phenotype in Msm unless expressed at higher than chromosomally expressed level. Conditions for induction of DinB2 expression are known (e.g., menaquinone, ∆MSMEG_2294, etc.) – why not try these conditions to induce physiologically relevant levels of chromosomally-expressed DinB2 and look for mutator phenotypes? I am still concerned that the in vivo mutagenic properties described here using overexpressed levels of DinB2/DinB3 may not be biologically relevant.

2) "Essential revision" point #3: the concern about the variation between WB in panels 1A vs 1E referred to the variable signal for the RpoD loading control. The weak RpoD signal (in contrast to other RpoD panels) may be outside the linear range of detection, drawing into question the accuracy of its use as a loading control and normalizing standard for the quantitation of RecA. Likewise, in panel A of Figure 1-supplement figure 1, the signal for the DinB2 and DinB3 proteins was very low (in contrast to other DinB2/DinB3 panels) and may not be in the linear range of detection. I am curious about the authors' views on why these panels have low signals while other panels do not, and the accuracy of the quantitation of these low signals.

3) "Other issues that should be addressed" point #4: figure 4, supplemental panels 1A-D, and figure 4B would benefit from more labels of molecular sizes for DNA products. For example, in figure 4 the authors state that products were in excess "~+60" in size, but the size markings only go up to 50 bp – it would greatly help the reader to see where the +60 product migrates to.

4) Pg. 6, lines 129-133: I realize that DinB1 is regulated by SigH, but can the authors exclude a role for chromosomally-expressed DinB1 in contributing to mutagenesis, similar to the arguments for RecA (SOS) and DnaE2?

5) Pg. 7, lines 158-162: The authors may want to relax/qualify their statement regarding DinB3 as they did not use a DinB3 active site mutation to confirm that mutations observed were in fact the result of DinB3 catalytic activity.

*Reviewer #2 (Recommendations for the authors):*

The authors have sufficiently addressed the reviewers' concerns. The authors explore the catalytic activity and mutagenic profile of DinB2. The intracellular function of DinB2 under physiological conditions remains elusive since only over-expression studies hint at its role in mutagenesis. Nevertheless, DinB2 may play a discernable role under certain conditions which upregulate its expression.

---

## [Author Response]

Essential revisions:1. Whilst the results are interesting, overexpression of the different DinBs is an artificial way of looking at their biological roles. Just because they increase mutation frequency when overproduced doesn't mean they act in DNA damage tolerance or mutagenesis when expressed at normal cellular levels from their native genomic loci. While this study is well done, it remains unclear whether DinB2 and/or DinB3 contribute to DNA damage tolerance and mutagenesis in Mycobacteria. Now that the authors have a unique mutation spectrum for DinB2 (when overexpressed), is it possible to search for that spectrum in response to DNA damage, and its disappearance in a strain lacking DinB2 function? This type of result would provide a strong argument that DinB2 is contributing to mutagenesis.

We agree that our work does not demonstrate that DinB2 has mutagenic activities in physiological conditions. This paper is a characterization of mutagenic proprieties of DinB2 in vivo and in vitro as well as homo-oligonucleotide runs. In our previous paper (PMID: 35918328), we found that the *dinB2* deletion does not cause a decrease of mutation frequency (substitutions or indels) and a focus on DinB2-specific mutations also did not reveal an effect. However, we note that the absence of an effect of a loss of function approach is only interpretable if the gene product is expressed in the conditions tested. There is evidence that *dinB2* expression is induced only under specific conditions. For instance, Barman et al., 2022 revealed that Menaquinone induces *dinB2* expression more than 10-fold. Although we cannot compare the absolute levels of DinB2 in our experiments to that study (due to their use of mRNA and our use of an epitope tagged protein) we have quantitated replicate western blots (as requested), which show a 10-fold increase of protein DinB2 level between – and + ATc conditions (Figure 1A), similar to the induction ratio in the literature. Thus, although we agree that our study determines the mutagenic potential of DinB2 without defining its physiologic role, the lack of a phenotype of the DinB2 deletion strain can be explained by lack of expression. We have added this reasoning (lines 66-72) to the manuscript introduction to justify our use of the inducible expression system.

2. The dependence on the catalytic activity of overproduced levels of DinB2 to impede growth is very reminiscent of work from Walker (Foti et al., Science (2012) 336:315-319) concluding that overproduced levels of *E. coli* DinB (Pol IV) kill by incorporating oxidized precursors into nascent DNA, leading to lethal numbers of dsDNA breaks. Have the authors tested whether killing induced by the overproduction of DinB2 can be suppressed by anaerobic growth or similar treatments to limit the production of oxidized DNA precursors?

We agree that the hypothesis by which the lethality induced by *dinB2* overexpression is due to genomic incorporation/excision of 8-oxoguanines (8-oxoG), as demonstrated in *E. coli* by Foti *et al.*, is very attractive. To test this possibility, we measured the effect of *dinB2* OE on bacterial growth and viability in *mutT1234* (8-oxoG degradation systems) and *mutYM12* (8-oxoG excision systems) mutants. We did not observe significant differences between genetic backgrounds suggesting that DinB2-dependent lethality is not mainly due to 8-oxoG genomic incorporation/excision.

This paragraph was added (lines 95-104) in addition to Figure1—figure supplement 2: “A study conducted in *E. coli* revealed that *dinB* overexpression toxicity is due to genomic incorporation and excision of 8-oxoguanines, leading to the formation of DNA double-strand breaks (Foti *et al.,* 2012). This is reminiscent with the ability of DinB2 to utilize 8-oxoguanine for DNA synthesis in vitro (Ordonez & Shuman, 2014). To test if DinB2 overexpression causes cell death in vivo because of genomic incorporation of 8-oxoguanines, we measured the impact of *mutT*s and *mutY/mutMs* deletions on DinB2-dependant lethality, systems involved in free 8-oxoguanines degradation and genomic 8-oxoguanines excision, respectively (Dupuy *et al.,* 2020). We did not observe a significant impact of the absence of these 8-oxoguanine processing systems on the effect of DinB2 (Figures S2A- D), indicating that DinB2-dependent lethality is not mainly due to genomic 8-oxoguanine incorporation.”

3. There are several concerns with the Western blots. First, can the authors comment on why panels 1A and 1E look so different? Second, quantitation of the blots would be helpful to gauge the expression levels of the DinBs? For example, was DinB3 expressed at the same level as DinB2, as a function of ATc concentration, and is growth impairment correlated with levels of DinB2 protein and not just ATc? Third, how many times were blots performed? Assuming each represents one of multiple reps their quantitation could indicate the average value +/- error (range or SD). Fourth, by eye, it looks like the DinB2L14F mutant (Figure 5) may be expressed at a slightly lower level than WT DinB2, but it has a more pronounced killing phenotype. Quantitation would support or refute the possibility that the ability to incorporate rNTPs impacts the growth defect caused by DinB2 overproduction. Finally, in Figure 6 was it confirmed that variable metal (Mg, Mn) concentrations failed to influence DinB2 levels by Western blot analysis?

We thank the reviewer for this suggestion. In response, we conducted new western blots (WB) of biological triplicates that are now included:

1. WB of DinB2 and DinB3 with different concentrations of ATC (Figure 1A).

2. WB of DinB2, DinB2^L107A^, and DinB2^L14F^ (Figure 1E and 5A).

3. WB of DinB2 in presence of variable concentrations of MnCl_2_ (Figure 6C).

We also performed quantifications of RecA WB presented in the 1^st^ version of the paper. The number of biological replicates and protein quantifications are now indicated under the picture of a representative blot.

4. The finding of kanR CFUs without any detectable mutations in the kan marker is worrisome and should be better discussed in the text. The same for sacB data in the supplementary material. The explanation given in lines 216-218 does not make sense. Markers 7G and 8G clearly are barely measuring any mutagenesis. The experiments wherein most of the supposed KanR revertants have no Kan mutation should either be removed from the manuscript or better discussed, because it is uncertain what they are measuring, therefore no conclusion can be drawn from them. For the Kan markers, one possible explanation is that translational frameshifts are occurring and allow residual growth of some of the cells. Gene amplification as seen in the lac system of Cairns and Foster in *E. coli* could also promote growth without actual mutations. Is the KanR phenotype of these colonies heritable and stable?

We agree with the reviewer that the 7G and 8G kanR reporters are not contributing to the study and we have removed them.

For the results obtained with SacB reporters, in the control strain (empty vector), a majority of suc^R^ clones do not have a *sacB* mutation, but their proportion is strongly induced after *dinB2* OE (because the frequency of FS in *sacB* increases). In addition to supporting the +1 and -1 FS activity of DinB2 in 6C and 9C runs, the primary purpose of the *sacB* data, because of its ability to report on longer insertion events due to its counter selectable function, is to exclude longer insertions that would be invisible in the kan system. For these two reasons, we have retained the *sacB* reporters in Figure 4—figure supplement 2.

5. Also, spontaneous mutagenesis should have been more precisely measured by using fluctuation analysis of larger sample sizes. In many instances, the results shown are the means of a few cultures with very large differences in mutant frequencies (several hundred-fold – e.g. Figures 4C,D and E, 5C and F, S3). Authors could discuss/explain their choice of statistical analysis and sample sizes.

We agree that fluctuation analysis is a very good way to compare spontaneous mutation frequencies between strains, cultivated through many generations because it corrects for jackpot effects of mutations arising early in the growth experiment. In our study, we measured mutation frequency after a short induction of DinBs using a 16h of ATC treatment (around 6 generations only), limiting the risk of jackpot effect. Because we observed a very strong induction of the mutation frequency (between 10- and 100-fold induction for most of the runs), and given the large number of strains/conditions tested in our study, we believe that the measure of mutation frequencies rather than fluctuation test (that needs several tens of replicates per condition) is appropriate.

Other issues that should be addressed:1. Title: Why are the authors are claiming "pleiotrpic" roles for these polymerases? What are the multiple phenotypes being observed, besides an effect on mutagenesis? The abstract is also not clear about this.

Pleiotropic has been deleted from the title.

2. Line 44: Error in the citation of Boshoff 2003.

Corrected.

3. Figures 1B and 1F: How OD600 values are measured? Are these very high densities really achievable in mycobacterial cultures (OD600 = 100)?

These growth curves, as stated in the methods, are continuous growth experiments to deduce an accurate doubling time. Cultures are diluted in fresh medium when they reach an OD_600_=1 and subsequent measured OD_600_ are corrected by the dilution factor for plotting. Thus, the Y axis does not represent a true OD, it is a calculated number. The procedure is described in the Methods section: “OD_600_ was measured for 48h. Cultures were diluted in fresh medium supplemented with ATc to a calculated OD_600_=0.001 when OD_600_ reached a value around 1 and measured OD_600_ was corrected by the dilution factor.”.

4. A simple change that would add a lot to the paper. In all gel images showing polymerase activity, an indicator of the molecular size of the relevant products discussed in the text would be of tremendous help to the readers. It is often confusing to follow what is being discussed about the length of slippage tracts without this aid.

Sizes (in nucleotides) of the relevant products are now indicated in the Figures depicting polymerase assays.

5. Lines 169-180: Why do the authors interpret products with an extra 8nt as the result of 8 rounds of slippage? Although this is possible, one slippage event could loop out more than one nucleotide, giving rise to the addition of more than one nucleotide. This idea is repeated later in the text when discussing figure 4.

We have added the following text:

“The heterogeneous size distribution of the slippage ladder is consistent with either of two scenarios: (i) multiple slippage cycles in which the primer 3’-OH end realigns backward on the template by a single nucleotide; or (ii) one or several cycles of backward realignment of the primer 3’-OH on the template by more than one nucleotide (the upper limit being the length of the template homooligomeric tract) followed by fill-in to the end of the homooligomeric tract.”

6. Line 180: "single cycle of forward slippage".

Corrected.

7. Line 249: Klenow polymerase domain.

Corrected.

8. Line 252: "DinB2 slippage in G or C…"

Corrected.

9. Line 259 and 260. The difference observed in the Kan marker 5G is not significant, as indicated in the figure.

The sentence was changed for: “However, *dinB2* overexpression enhanced -1 FS frequency in the 4G run by 100-fold and non-statistically significant increase of +1 FS frequency was observed in the 5G run.”

10. Lines 333 to 334: "Magnesium did not impact the relatively high level of slippage on the G6 template that was seen in the presence of ddATP." This is not clear, given that the experiment shows a clear effect of Mn ions. Would the point be that increasing concentrations of Mn do not enhance slippage significantly after the transition seen at 0.5 mM?

The metal concentration dependence of the shift in slippage behavior is relevant insofar as the standard assays for slippage contained 5 mM manganese (which we now state explicitly in Results text) and the experiment presented makes clear that the shift in the presence of 5 mM magnesium occurs at lower concentrations of manganese (0.5 mM).

11. Figures 6E and 6F: indicate which reporters are being used. 4T, 5T?

Done.

12. The authors state in Methods that at least 3 cultures were sequenced for each mutagenesis experiment, but how many of the sequences came from each culture?

It differs depending on the experiment. the sentence “(between 2 and 8 clones per independent culture were sequenced)” was added to the methods section.

13. Figure S2 shows results for only DinB2, not DinB2 and DinB3, as stated in the text (lines 104-107).

DinB3 data was added to Figure 2—figure supplement 1.

14. Did the authors examine DinB3 replication on some of the same templates as DinB2, and if so did it also slip? If not, this would further support the conclusion that slippage is the cause of the frameshifts.

Although we agree this is an interesting question, the in vitro frameshifting activity of DinB3 was not examined here.

15. Line 250: was this slippage or misincorporation?

We cannot distinguish, but the noted activity of PolI is much weaker than DinB2.

16. Both DinB2 and DinB3 over-expression result in the induction of RecA. The extent of upregulation of the SOS response is not readily discerned from the RecA immunoblot. Do the authors have any associated transcriptional data to investigate any differences?Figure S2: the DinB3 over-expression in the dnaE2 and recA deletion mutants is missing but referred to in the text. This is an important control and the panel would be good to include.

Data of substitution mutagenesis after *dinB3* OE in Δ*recA* and Δ*dnaE2* mutants were added in Figure 2—figure supplement 1.

17. Manganese supplementation of the culture medium does not increase the frequency of slippage in the induced tet-dinB2 strain and probably does not significantly affect that in the control strain either. A good additional control would have been the mutant lacking dnaE2 and dinB1-3 used in the Dupuy et al. (2022) study.

We are not clear on what the reviewer is asking. The lack of effect of Mn in the control strain would seem to exclude the involvement of any of the polymerases and therefore the null strains would not be informative.

18. This study demonstrates the role of DinB2-mediated mutagenesis in vivo. It can incorporate ribonucleotides as well as dNTPs. The ratio of these could affect its in vivo activity. One wonders whether the over-expression of DinB2 and the DinB2-L14F (steric mutant) result in altered mutation frequencies when plated from starved cells.

We agree with this point by the reviewer and exploring the role of DinB2 in starved cells will be part of a future study.

19. Figure 1A vs Figure 5A: why is the DinB immunoblotting so weak in 1A?

New western-blots were performed in triplicate and band intensity was quantified as noted above and in the figures.

20. Line 423: it would be informative to compare homo-oligonucleotide runs in bacteria such as *E. coli* and *B. subtilis* to get a better sense of how common such tracts are between different species.

We agree that this is an interesting bioinformatic question and we agree that our study and others may stimulate such a comparison.

[Editors' note: further revisions were suggested prior to acceptance, as described below.]

The manuscript has been improved but there are some remaining issues that need to be addressed, as outlined below:Reviewers agreed that the revised manuscript has been substantively improved, there is however one primary concern remaining. This relates to ascribing a function for DinB2 in mutagenesis, particularly the conditions under which such activity manifests. As a first request, please consider revising your discussion, and appending text in the abstract to specifically address this limitation. Refer to the detailed comments from the reviewers to get a better sense of this concern, particularly point 1 from Reviewer 1.Secondly, the following points need addressing, also through revision of the manuscript to acknowledge any limitations and clarify data presentation:1. The concern about the variation between western blots in panels 1A vs 1E referred to the variable signal for the RpoD loading control. The weak RpoD signal (in contrast to other RpoD panels) may be outside the linear range of detection, drawing into question the accuracy of its use as a loading control and normalizing standard for the quantitation of RecA. Likewise, in panel A of Figure 1-supplement figure 1, the signal for the DinB2 and DinB3 proteins was very low (in contrast to other DinB2/DinB3 panels) and may not be in the linear range of detection. Can you the authors explain this, specifically with respect to the accuracy of the quantitation of these low signals.

We presume these comments relate to RpoB, the loading control used, not RpoD. Regarding the linear range of detection, as stated in the methods, these blots were imaged on an iBright FL1000, which has a large dynamic range and alerts the user to bands that are outside of the dynamic range of the instrument. We can confirm that this quantitation was done in the linear range and have added this detail to the methods. Regarding the statement that the band intensity is different between gels, we emphasize that variable band intensity between blots is intrinsic to the technique of immunoblotting. Variation in transfer efficiency, antibody variability, age of chemiluminescent reagent, and many other variable affect band intensities between blots. This variability is exactly the reason that we (and others) employ loading controls as the most rigorous way to normalize these variables and provide an internal control for quantitation on each blot, which we have done for 3 replicate blots to derive the numbers that are presented.

2. Figure 4, supplemental panels 1A-D, and figure 4B would benefit from more labels of molecular sizes for DNA products. For example, in figure 4 the authors state that products were in excess "~+60" in size, but the size markings only go up to 50 bp – it would greatly help the reader to see where the +60 product migrates to.

Unfortunately, we are unable to provide additional markers on these gels. We have altered the text to say >50 since the products are above the 50bp marker.

3. Pg. 6, lines 129-133: DinB1 is regulated by SigH, but can the authors exclude a role for chromosomally-expressed DinB1 in contributing to mutagenesis, similar to the arguments for RecA (SOS) and DnaE2?

We thank for reviewer for this comment. The mutation spectrum catalyzed by DinB2 is most similar to DnaE2 in that it appears in multiple codons within the RRDR. As noted in the text, the Rif^R^ mutations catalyzed by DinB1 (and DinB3, as shown here) are focused on a single substitution with a single codon (His442 CAC>CGC). Although this mutation is part of the DinB2 mutation spectrum (see figure 4D) it is a small fraction and therefore it is unlikely that DinB1 is responsible for the DinB2 mutation picture. We cannot exclude that DinB1 contributes to the small number of His442 CAC>CGC mutations observed.

4. Pg. 7, lines 158-162: The authors may want to relax/qualify their statement regarding DinB3 as they did not use a DinB3 active site mutation to confirm that mutations observed were in fact the result of DinB3 catalytic activity.

We have added the following sentence to the discussion:

“This DinB2 effect is dependent on its polymerase activity, whereas we did not express a DinB3 active site mutant.”